# Data Amplification: A Unified and Competitive Approach to Property Estimation

**Yi HAO**
Dept. of Electrical and Computer Engineering
University of California, San Diego
La Jolla, CA 92093
yih179@eng.ucsd.edu

**Alon Orlitsky**
Dept. of Electrical and Computer Engineering
University of California, San Diego
La Jolla, CA 92093
alon@eng.ucsd.edu

**Ananda T. Suresh**
Google Research, New York
New York, NY 10011
theertha@google.com

**Yihong Wu**
Dept. of Statistics and Data Science
Yale University
New Haven, CT 06511
yihong.wu@yale.edu

## Abstract

Estimating properties of discrete distributions is a fundamental problem in statistical learning. We design the first unified, linear-time, competitive, property estimator that for a wide class of properties and for all underlying distributions uses just $2n$ samples to achieve the performance attained by the empirical estimator with $n\sqrt{\log n}$ samples. This provides off-the-shelf, distribution-independent, "amplification" of the amount of data available relative to common-practice estimators.

We illustrate the estimator's practical advantages by comparing it to existing estimators for a wide variety of properties and distributions. In most cases, its performance with $n$ samples is even as good as that of the empirical estimator with $n \log n$ samples, and for essentially all properties, its performance is comparable to that of the best existing estimator designed specifically for that property.

## 1 Distribution Properties

Let $D_{\mathcal{X}}$ denote the collection of distributions over a countable set $\mathcal{X}$ of finite or infinite cardinality $k$. A distribution *property* is a mapping $f : D_{\mathcal{X}} \to \mathbb{R}$. Many applications call for estimating properties of an unknown distribution $p \in D_{\mathcal{X}}$ from its samples. Often these properties are *additive*, namely can be written as a sum of functions of the probabilities. Symmetric additive properties can be written as

$$f(p) \overset{\text{def}}{=} \sum_{x \in \mathcal{X}} f(p_x),$$

and arise in many biological, genomic, and language-processing applications:

**Shannon entropy** $\sum_{x \in \mathcal{X}} p_x \log \frac{1}{p_x}$, where throughout the paper $\log$ is the natural logarithm, is the fundamental information measure arising in a variety of applications [1].

**Normalized support size** $\sum_{x \in \mathcal{X}} \frac{1}{k} \mathbb{1}_{p_x > 0}$ plays an important role in population [2] and vocabulary size estimation [3].

**Normalized support coverage** $\sum_{x \in \mathcal{X}} \frac{1 - e^{-m p_x}}{m}$ is the normalized expected number of distinct elements observed upon drawing $\text{Poi}(m)$ independent samples, it arises in ecological [4], genomic [5], and database studies [6].

**Power sum** $\sum_{x\in\mathcal{X}} p_x^a$, arises in Rényi entropy [7], Gini impurity [8], and related diversity measures.

**Distance to uniformity** $\sum_{x\in\mathcal{X}} \left|p_x - \frac{1}{k}\right|$, appears in property testing [9].

More generally, non-symmetric additive properties can be expressed as

$$f(p) \stackrel{\text{def}}{=} \sum_{x\in\mathcal{X}} f_x(p_x),$$

for example distances to a given distribution, such as:

**L1 distance** $\sum_{x\in\mathcal{X}} |p_x - q_x|$, the $L_1$ distance of the unknown distribution $p$ from a given distribution $q$, appears in hypothesis-testing errors [10].

**KL divergence** $\sum_{x\in\mathcal{X}} p_x \log \frac{p_x}{q_x}$, the KL divergence of the unknown distribution $p$ from a given distribution $q$, reflects the compression [1] and prediction [11] degradation when estimating $p$ by $q$.

Given one of these, or other, properties, we would like to estimate its value based on samples from an underlying distribution.

## 2 Recent Results

In the common property-estimation setting, the unknown distribution $p$ generates $n$ i.i.d. samples $X^n \sim p^n$, which in turn are used to estimate $f(p)$. Specifically, given property $f$, we would like to construct an *estimator* $\hat{f} : \mathcal{X}^* \to \mathbb{R}$ such that $\hat{f}(X^n)$ is as close to $f(p)$ as possible. The standard *estimation loss* is the expected squared loss

$$\mathbb{E}_{X^n \sim p^n} \left( \hat{f}(X^n) - f(p) \right)^2.$$

Generating exactly $n$ samples creates dependence between the number of times different symbols appear. To avoid these dependencies and simplify derivations, we use the well-known *Poisson sampling* [12] paradigm. We first select $N \sim \text{Poi}(n)$, and then generate $N$ independent samples according to $p$. This modification does not change the statistical nature of the estimation problem since a Poisson random variables is exponentially concentrated around its mean. Correspondingly the estimation loss is

$$L_{\hat{f}}(p, n) \stackrel{\text{def}}{=} \mathbb{E}_{N\sim\text{Poi}(n)} \left[ \mathbb{E}_{X^N \sim p^N} \left( \hat{f}(X^N) - f(p) \right)^2 \right].$$

For simplicity, let $N_x$ be the number of occurrences of symbol $x$ in $X^n$. An intuitive estimator is the plug-in *empirical estimator* $f^E$ that first uses the $N$ samples to estimate $p_x = N_x/N$ and then estimates $f(p)$ as

$$f^E(X^N) \stackrel{\text{def}}{=} \begin{cases} \sum_{x\in\mathcal{X}} f_x\left(\frac{N_x}{N}\right) & N > 0, \\ 0 & N = 0. \end{cases}$$

Given an error tolerance parameter $\delta > 0$, the $(\delta, p)$-*sample complexity* of an estimator $\hat{f}$ in estimating $f(p)$ is the smallest number of samples $n$ allowing for estimation loss smaller than $\delta$,

$$n_{\hat{f}}(\delta, p) \stackrel{\text{def}}{=} \min_{n\in\mathbb{N}} \{L_{\hat{f}}(p, n) < \delta\}.$$

Since $p$ is unknown, the common *min-max* approach considers the worst case $(\delta, p)$-sample complexity of an estimator $\hat{f}$ over all possible $p$,

$$n_{\hat{f}}(\delta) \stackrel{\text{def}}{=} \max_{p\in D_{\mathcal{X}}} n_{\hat{f}}(\delta, p).$$

Finally, the estimator minimizing $n_{\hat{f}}(\delta)$ is called the *min-max estimator* of property $f$, denoted $f^{\text{M}}$. It follows that $n_{f^M}(\delta)$ is the smallest Poisson parameter $n$, or roughly the number of samples, needed for any estimator $\hat{f}$ to estimate $f(p)$ to estimation loss $\delta$ for all $p$.

There has been a significant amount of recent work on property estimation. In particular, it was shown that for all seven properties mentioned earlier, $f^M$ improves the sample complexity by a logarithmic factor compared to $f^E$. For example, for Shannon entropy [13], normalized support size [14], normalized support coverage [15], and distance to uniformity [16], $n_{f^E}(\delta) = \Theta_\delta(k)$ while $n_{f^M}(\delta) = \Theta_\delta(k/\log k)$. Note that for normalized support size, $D_{\mathcal{X}}$ is typically replaced by $D_k := \{p \in D_{\mathcal{X}} : p_x \geq 1/k, \forall x \in \mathcal{X}\}$, and for normalized support coverage, $k$ is replaced by $m$.

## 3 New Results

While the results already obtained are impressive, they also have some shortcomings. Recent state-of-the-art estimators are designed [13, 14, 16] or analyzed [15, 19] to estimate each individual property. Consequently these estimators cover only few properties. Second, estimators proposed for more general properties [15, 20] are limited to symmetric properties and are not known to be computable in time linear in the sample size. Last but not least, by design, min-max estimators are optimized for the "worst" distribution in a class. In practice, this distribution is often very different, and frequently much more complex, than the actual underlying distribution. This "pessimistic" worst-case design results in sub-optimal estimation, as born by both the theoretical and experimental results.

In Section 6, we design an estimator $f^*$ that addresses all these issues. It is *unified* and applies to a wide range of properties, including all previously-mentioned properties ($a > 1$ for power sums) and all *Lipschitz properties* $f$ where each $f_x$ is Lipschitz. It can be computed in *linear-time* in the sample size. It is *competitive* in that it is guaranteed to perform well not just for the worst distribution in the class, but for each and every distribution. It *"amplifies"* the data in that it uses just $\text{Poi}(2n)$ samples to approximate the performance of the empirical estimator with $\text{Poi}(n\sqrt{\log n})$ samples *regardless of the underlining distribution* $p$, thereby providing an off-the-shelf, distribution-independent, "amplification" of the amount of data available relative to the estimators used by many practitioners. As we show in Section 8, it also works well in practice, outperforming existing estimator and often working as well as the empirical estimator with even $n \log n$ samples.

For a more precise description, let $o(1)$ represent a quantity that vanishes as $n \to \infty$ and write $a \lesssim b$ for $a \leq b(1 + o(1))$. Suppressing small $\epsilon$ for simplicity first, we show that

$$L_{f^*}(p, 2n) \lesssim L_{f^E}(p, n\sqrt{\log n}) + o(1),$$

where the first right-hand-side term relates the performance of $f^*$ with $2n$ samples to that of $f^E$ with $n\sqrt{\log n}$ samples. The second term adds a small loss that diminishes at a rate independent of the support size $k$, and for fixed $k$ decreases roughly as $1/n$. Specifically, we prove,

**Theorem 1.** *For every property $f$ satisfying the smoothness conditions in Section 5, there is a constant $C_f$ such that for all $p \in D_{\mathcal{X}}$ and all $\epsilon \in (0, \frac{1}{2})$,*

$$L_{f^*}(p, 2n) \leq \left(1 + \frac{3}{\log^\epsilon n}\right) L_{f^E}\left(p, n \log^{\frac{1}{2} - \epsilon} n\right) + C_f \min\left\{\frac{k}{n}\log^\epsilon n + \tilde{\mathcal{O}}\left(\frac{1}{n}\right), \frac{1}{\log^\epsilon n}\right\}.$$

The $\tilde{\mathcal{O}}$ reflects a multiplicative polylog($n$) factor unrelated to $k$ and $p$. Again, for normalized support size, $D_{\mathcal{X}}$ is replaced by $D_k$, and we also modify $f^*$ as follows: if $k > n$, we apply $f^*$, and if $k \leq n$, we apply the corresponding min-max estimator [14]. However, for experiments shown in Section 8, the original $f^*$ is used without such modification. In Section 7, we note that for several properties, the second term can be strengthened so that it does not depend on $\epsilon$.

## 4 Implications

Theorem 1 has three important implications.

**Data amplification** Many modern applications, such as those arising in genomics and natural-language processing, concern properties of distributions whose support size $k$ is comparable to or even larger than the number of samples $n$. For these properties, the estimation loss of the empirical estimator $f^E$ is often much larger than $1/\log^\epsilon n$, hence the proposed estimator, $f^*$, yields a much better estimate whose performance parallels that of $f^E$ with $n\sqrt{\log n}$ samples. This allows us to amplify the available data by a factor of $\sqrt{\log n}$ regardless of the underlying distribution.

Note however that for some properties $f$, when the underlying distributions are limited to a fixed small support size, $L_{f^E}(p, n) = \Theta(1/n) \ll 1/\log^\epsilon n$. For such small support sizes, $f^*$ may not improve the estimation loss.

**Unified estimator**   Recent works either prove efficacy results individually for each property [13, 14, 16], or are not known to be computable in linear time [15, 20].

By contrast, $f^*$ is a linear-time estimator well for all properties satisfying simple Lipschitz-type and second-order smoothness conditions. All properties described earlier: Shannon entropy, normalized support size, normalized suppport coverage, power sum, $L_1$ distance and KL divergence satisfy these conditions, and $f^*$ therefore applies to all of them.

More generally, recall that a property $f$ is Lipschitz if all $f_x$ are Lipschitz. It can be shown, e.g. [21], that with $\mathcal{O}(k)$ samples, $f^E$ approximates a $k$-element distribution to a constant $L_1$ distance, and hence also estimates any Lipschitz property to a constant loss. It follows that $f^*$ estimates any Lipschitz property over a distribution of support size $k$ to constant estimation loss with $\mathcal{O}(k/\sqrt{\log k})$ samples. This provides the first general sublinear-sample estimator for all Lipschitz properties.

**Competitive optimality**   Previous results were geared towards the estimator's worst estimation loss over all possible distributions. For example, they derived estimators that approximate the distance to uniformity of any $k$-element distribution with $\mathcal{O}(k/\log k)$ samples, and showed that this number is optimal as for some distribution classes estimating this distance requires $\Omega(k/\log k)$ samples.

However, this approach may be too pessimistic. Distributions are rarely maximally complex, or are hardest to estimate. For example, most natural scenes have distinct simple patterns, such as straight lines, or flat faces, hence can be learned relatively easily.

More concretely, consider learning distance to uniformity for the collection of distributions with entropy bounded by $\log \log k$. It can be shown that for sufficiently large $k$, $f^E$ can learn distance to uniformity to constant estimation loss using $\mathcal{O}((\log k)^{\Theta(1)})$ samples. Theorem 1 therefore shows that the distance to uniformity can be learned to constant estimation loss with $\mathcal{O}((\log k)^{\Theta(1)}/\sqrt{\log \log k})$ samples. (In fact, without even knowing that the entropy is bounded.) By contrast, the original min-max estimator results would still require the much larger $\Omega(k/\log k)$ samples.

The rest of the paper is organized as follows. Section 5 describes mild smoothness conditions satisfied by many natural properties, including all those mentioned above. Section 6 describes the estimator's explicit form and some intuition behind its construction and performance. Section 7 describes two improvements of the estimator addressed in the supplementary material. Lastly, Section 8 describes various experiments that illustrate the estimator's power and competitiveness. For space considerations, we relegate all the proofs to the supplemental material.

## 5   Smooth properties

Many natural properties, including all those mentioned in the introduction satisfy some basic smoothness conditions. For $h \in (0, 1]$, consider the Lipschitz-type parameter

$$\ell_f(h) \stackrel{\text{def}}{=} \max_x \max_{u,v \in [0,1]:\max\{u,v\} \geq h} \frac{|f_x(u) - f_x(v)|}{|u - v|},$$

and the second-order smoothness parameter, resembling the modulus of continuity in approximation theory [17, 18],

$$\omega_f^2(h) \stackrel{\text{def}}{=} \max_x \max_{u,v \in [0,1]:|u-v| \leq 2h} \left\{ \left| \frac{f_x(u) + f_x(v)}{2} - f_x\left(\frac{u+v}{2}\right) \right| \right\}.$$

We consider properties $f$ satisfying the following conditions: (1) $\forall x \in \mathcal{X}$, $f_x(0) = 0$; (2) $\ell_f(h) \leq$ polylog$(1/h)$ for $h \in (0, 1]$; (2) $\omega_f^2(h) \leq S_f \cdot h$ for some absolute constant $S_f$.

Note that the first condition, $f_x(0) = 0$, entails no loss of generality. The second condition implies that $f_x$ is continuous over $[0, 1]$, and in particular right continuous at 0 and left-continuous at 1. It is easy to see that continuity is also essential for consistent estimation. Observe also that these conditions are more general than assuming that $f_x$ is Lipschitz, as can be seen for entropy where $f_x = x \log x$, and that all seven properties described earlier satisfy these three conditions. Finally, to ensure that $L_1$ distance satisfies these conditions, we let $f_x(p_x) = |p_x - q_x| - q_x$.

# 6 The Estimator $f^*$

Given the sample size $n$, define an *amplification parameter* $t > 1$, and let $N'' \sim \text{Poi}(nt)$ be the amplified sample size. Generate a sample sequence $X^{N''}$ independently from $p$, and let $N''_x$ denote the number of times symbol $x$ appeared in $X^{N''}$. The empirical estimate of $f(p)$ with $\text{Poi}(nt)$ samples is then

$$f^E(X^{N''}) = \sum_{x \in \mathcal{X}} f_x \left( \frac{N''_x}{N''} \right).$$

Our objective is to construct an estimator $f^*$ that approximates $f^E(X^{N''})$ for large $t$ using just $\text{Poi}(2n)$ samples.

Since $N''$ sharply concentrates around $nt$, we can show that $f^E(X^{N''})$ can be approximated by the *modified empirical estimator*,

$$f^{ME}(X^{N''}) \overset{\text{def}}{=} \sum_{x \in \mathcal{X}} f_x \left( \frac{N''_x}{nt} \right),$$

where $f_x(p) \overset{\text{def}}{=} f_x(1)$ for all $p > 1$ and $x \in \mathcal{X}$.

Since large probabilities are easier to estimate, it is natural to set a threshold parameter $s$ and rewrite the modified estimator as a separate sum over small and large probabilities,

$$f^{ME}(X^{N''}) = \sum_{x \in \mathcal{X}} f_x \left( \frac{N''_x}{nt} \right) \mathbb{1}_{p_x \leq s} + \sum_{x \in \mathcal{X}} f_x \left( \frac{N''_x}{nt} \right) \mathbb{1}_{p_x > s}.$$

Note however that we do not know the exact probabilities. Instead, we draw two independent sample sequences $X^N$ and $X^{N'}$ from $p$, each of an independent $\text{Poi}(n)$ size, and let $N_x$ and $N'_x$ be the number of occurrences of $x$ in the first and second sample sequence respectively. We then set a *small/large-probability threshold* $s_0$ and classify a probability $p_x$ as large or small according to $N'_x$:

$$f^{ME}_S(X^{N''}, X^{N'}) \overset{\text{def}}{=} \sum_{x \in \mathcal{X}} f_x \left( \frac{N''_x}{nt} \right) \mathbb{1}_{N'_x \leq s_0}$$

is the *modified small-probability empirical estimator*, and

$$f^{ME}_L(X^{N''}, X^{N'}) \overset{\text{def}}{=} \sum_{x \in \mathcal{X}} f_x \left( \frac{N''_x}{nt} \right) \mathbb{1}_{N'_x > s_0}$$

is the *modified large-probability empirical estimator*. We rewrite the modified empirical estimator as

$$f^{ME}(X^{N''}) = f^{ME}_S(X^{N''}, X^{N'}) + f^{ME}_L(X^{N''}, X^{N'}).$$

Correspondingly, we express our estimator $f^*$ as a combination of small- and large-probability estimators,

$$f^*(X^N, X^{N'}) \overset{\text{def}}{=} f^*_S(X^N, X^{N'}) + f^*_L(X^N, X^{N'}).$$

The *large-probability estimator* approximates $f^{ME}_L(X^{N''}, X^{N'})$ as

$$f^*_L(X^N, X^{N'}) \overset{\text{def}}{=} f^{ME}_L(X^N, X^{N'}) = \sum_{x \in \mathcal{X}} f_x \left( \frac{N_x}{nt} \right) \mathbb{1}_{N'_x > s_0}.$$

Note that we replaced the length-$\text{Poi}(nt)$ sample sequence $X^{N''}$ by the independent length-$\text{Poi}(n)$ sample sequence $X^N$. We can do so as large probabilities are well estimated from fewer samples.

The *small-probability estimator* $f^*_S(X^N, X^{N'})$ approximates $f^{ME}_S(X^{N''}, X^{N'})$ and is more involved. We outline its construction below and details can be found in Section 8 of the supplemental material. The expected value of $f^{ME}$ for the small probabilities is

$$\mathbb{E}[f^{ME}_S(X^{N''}, X^{N'})] = \sum_{x \in \mathcal{X}} \mathbb{E}[\mathbb{1}_{N_x \leq s_0}] \mathbb{E}\left[ f_x \left( \frac{N''_x}{nt} \right) \right].$$

Let $\lambda_x \overset{\text{def}}{=} np_x$ be the expected number of times symbol $x$ will be observed in $X^N$, and define

$$g_x(v) \overset{\text{def}}{=} f_x\left(\frac{v}{nt}\right)\left(\frac{t}{t-1}\right)^v.$$

Then

$$\mathbb{E}\left[f_x\left(\frac{N_x''}{nt}\right)\right] = \sum_{v=0}^{\infty} e^{-\lambda_x t}\frac{(\lambda_x t)^v}{v!}f_x\left(\frac{v}{nt}\right) = e^{-\lambda_x}\sum_{v=1}^{\infty} e^{-\lambda_x(t-1)}\frac{(\lambda_x(t-1))^v}{v!}g_x(v).$$

As explained in Section 8.1 of the supplemental material, the sum beyond a *truncation threshold*

$$u_{\max} \overset{\text{def}}{=} 2s_0 t + 2s_0 - 1$$

is small, hence it suffices to consider the truncated sum

$$e^{-\lambda_x}\sum_{v=1}^{u_{\max}} e^{-\lambda_x(t-1)}\frac{(\lambda_x(t-1))^v}{v!}g_x(v).$$

Applying the *polynomial smoothing technique* in [22], Section 8.2 of the supplemental material approximates the above summation by

$$e^{-\lambda_x}\sum_{v=1}^{\infty} h_{x,v}\lambda_x^v,$$

where

$$h_{x,v} = (t-1)^v\sum_{u=1}^{(u_{\max}\wedge v)} \frac{g_x(u)(-1)^{v-u}}{(v-u)!u!}\left(1 - e^{-r}\sum_{j=0}^{v+u}\frac{r^j}{j!}\right),$$

and

$$r \overset{\text{def}}{=} 10s_0 t + 10s_0.$$

Observe that $1 - e^{-r}\sum_{j=0}^{v+u}\frac{r^j}{j!}$ is the tail probability of a $\text{Poi}(r)$ distribution that diminishes rapidly beyond $r$. Hence $r$ determines which summation terms will be attenuated, and serves as a *smoothing parameter*.

An unbiased estimator of $e^{-\lambda_x}\sum_{v=1}^{\infty} h_{x,v}\lambda_x^v$ is

$$\sum_{v=1}^{\infty} h_{x,v}v! \cdot \mathbb{1}_{N_x=v} = h_{x,N_x}\cdot N_x!.$$

Finally, the small-probability estimator is

$$f_S^*(X^N, X^{N'}) \overset{\text{def}}{=} \sum_{x\in\mathcal{X}} h_{x,N_x}\cdot N_x!\cdot \mathbb{1}_{N_x'\leq s_0}.$$

## 7  Extensions

In Theorem 1, for fixed $n$, as $\epsilon \to 0$, the final slack term $1/\log^\epsilon n$ approaches a constant. For certain properties it can be improved. For normalized support size, normalized support coverage, and distance to uniformity, a more involved estimator improves this term to

$$C_{f,\gamma}\min\left\{\frac{k}{n\log^{1-\epsilon} n} + \frac{1}{n^{1-\gamma}}, \frac{1}{\log^{1+\epsilon} n}\right\},$$

for any fixed constant $\gamma \in (0, 1/2)$.

For Shannon entropy, correcting the bias of $f_L^*$ [23] and further dividing the probability regions, reduces the slack term even more, to

$$C_{f,\gamma}\min\left\{\frac{k^2}{n^2\log^{2-\epsilon} n} + \frac{1}{n^{1-\gamma}}, \frac{1}{\log^{2+2\epsilon} n}\right\}.$$

Finally, the theorem compares the performance of $f^*$ with $2n$ samples to that of $f^E$ with $n\sqrt{\log n}$ samples. As shown in the next section, the performance is often comparable to that of $n\log n$ samples. It would be interesting to prove a competitive result that enlarges the amplification to $n\log^{1-\epsilon} n$ or even $n\log n$. This would be essentially the best possible as it can be shown that for the symmetric properties mentioned in the introduction, amplification cannot exceed $\mathcal{O}(n\log n)$.

# 8 Experiments

We evaluated the new estimator $f^*$ by comparing its performance to several recent estimators [13–15, 22, 27]. To ensure robustness of the results, we performed the comparisons for all the symmetric properties described in the introduction: entropy, support size, support coverage, power sums, and distance to uniformity. For each property, we considered six underlying distributions: uniform, Dirichlet-drawn-, Zipf, binomial, Poisson, and geometric. The results for the first three properties are shown in Figures 1–3, the plots for the final two properties can be found in Section 9 of the supplemental material. *For nearly all tested properties and distributions, $f^*$ achieved state-of-the-art performance.*

As Theorem 1 implies, for all five properties, with just $n$ (not even $2n$) samples, $f^*$ performed as well the empirical estimator $f^E$ with roughly $n\sqrt{\log n}$ samples. Interestingly, in most cases $f^*$ performed even better, similar to $f^E$ with $n \log n$ samples.

Relative to previous estimators, depending on the property and distribution, different previous estimators were best. But in essentially all experiments, $f^*$ was either comparable or outperformed the best previous estimator. The only exception was PML that attempts to smooth the estimate, hence performed better on uniform, and near-uniform Dirichlet-drawn distributions for several properties.

Two additional advantages of $f^*$ may be worth noting. First, underscoring its competitive performance for each distribution, the more skewed the distribution the better is its relative efficacy. This is because most other estimators are optimized for the worst distribution, and work less well for skewed ones.

Second, by its simple nature, the empirical estimator $f^E$ is very stable. Designed to emulate $f^E$ for more samples, $f^*$ is therefore stable as well. Note also that $f^E$ is not always the best estimator choice. For example, it always underestimates the distribution's support size. Yet even for normalized support size, Figure 2 shows that $f^*$ outperforms other estimators including those designed specifically for this property (except as above for PML on near-uniform distributions).

The next subsection describes the experimental settings. Additional details and further interpretation of the observed results can be found in Section 9 of the supplemental material.

### Experimental settings

We tested the five properties on the following distributions: uniform distribution; a distribution randomly generated from Dirichlet prior with parameter 2; Zipf distribution with power 1.5; Binomial distribution with success probability 0.3; Poisson distribution with mean 3,000; geometric distribution with success probability 0.99.

With the exception of normalized support coverage, all other properties were tested on distributions of support size $k = 10,000$. The Geometric, Poisson, and Zipf distributions were truncated at $k$ and re-normalized. The number of samples, $n$, ranged from 1,000 to 100,000, shown logarithmically on the horizontal axis. Each experiment was repeated 100 times and the reported results, shown on the vertical axis, reflect their mean squared error (MSE).

We compared the estimator's performance with $n$ samples to that of four other recent estimators as well as the empirical estimator with $n$, $n\sqrt{\log n}$, and $n \log n$ samples. We chose the amplification parameter $t$ as $\log^{1-\alpha} n + 1$, where $\alpha \in \{0.0, 0.1, 0.2, ..., 0.6\}$ was selected based on independent data, and similarly for $s_0$. Since $f^*$ performed even better than Theorem 1 guarantees, $\alpha$ ended up between 0 and 0.3 for all properties, indicating amplification even beyond $n\sqrt{\log n}$. The graphs denote $f^*$ by NEW, $f^E$ with $n$ samples by Empirical, $f^E$ with $n\sqrt{\log n}$ samples by Empirical+, $f^E$ with $n \log n$ samples by Empirical++, the pattern maximum likelihood estimator in [15] by PML, the Shannon-entropy estimator in [27] by JVHW, the normalized-support-size estimator in [14] and the entropy estimator in [13] by WY, and the smoothed Good-Toulmin Estimator for normalized support coverage estimation [22], slightly modified to account for previously-observed elements that may appear in the subsequent sample, by SGT.

While the empirical and the new estimators have the same form for all properties, as noted in the introduction, the recent estimators are property-specific, and each was derived for a subset of the properties. In the experiments we applied these estimators to all the properties for which they were derived. Also, additional estimators [28–34] for various properties were compared in [13, 14, 22, 27] and found to perform similarly to or worse than recent estimators, hence we do not test them here.

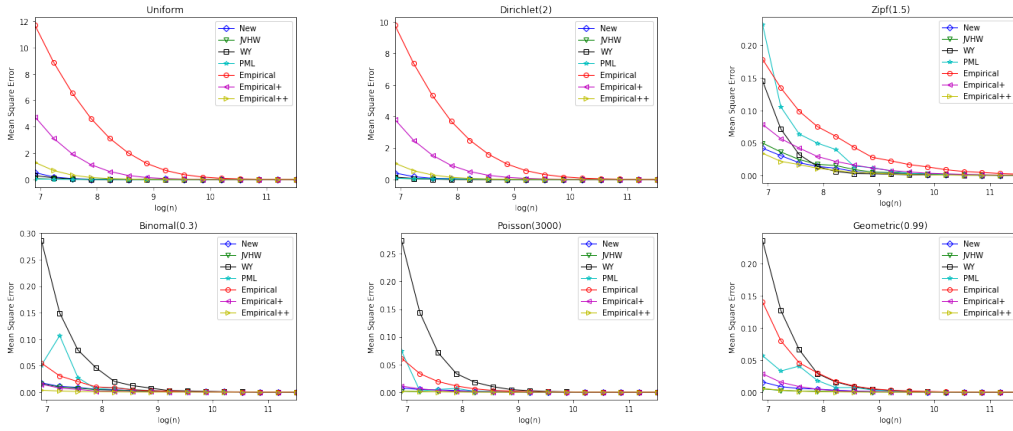

Figure 1: Shannon Entropy

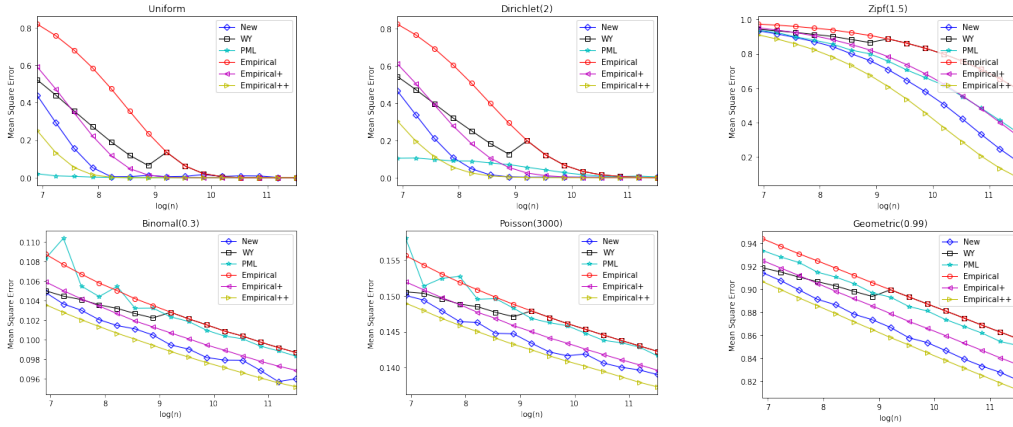

Figure 2: Normalized Support Size

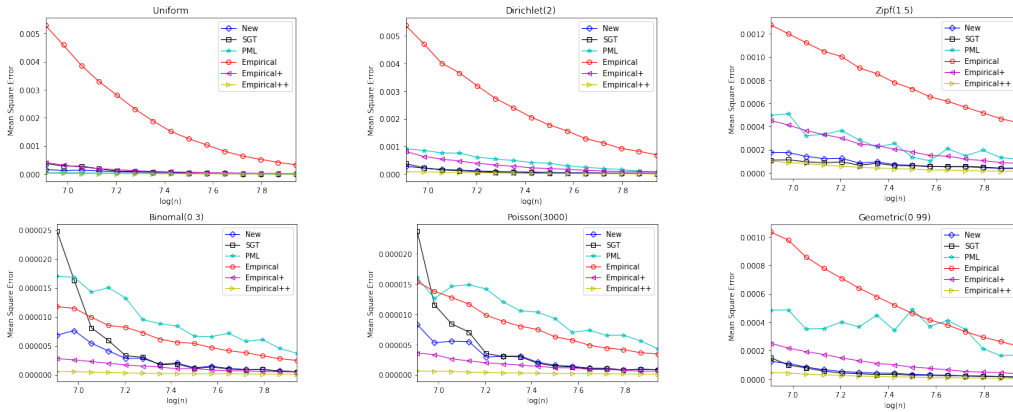

Figure 3: Normalized Support Coverage

## 9  Conclusion

In this paper, we considered the fundamental learning problem of estimating properties of discrete distributions. The best-known distribution-property estimation technique is the "empirical estimator" that takes the data's empirical frequency and plugs it in the property functional. We designed a general estimator that for a wide class of properties, uses only $n$ samples to achieve the same accuracy as the plug-in estimator with $n\sqrt{\log n}$ samples. This provides an off-the-shelf method for *amplifying* the data available relative to traditional approaches. For all the properties and distributions we have tested, the proposed estimator performed as well as the best estimator(s). A meaningful future research direction would be to verify the optimality of our results: the amplification factor $\sqrt{\log n}$ and the slack terms. There are also several important properties that are not included in our paper, for example, Rényi entropy [35] and the generalized distance to uniformity [36, 37]. It would be interesting to determine whether *data amplification* could be obtained for these properties as well.

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
