[Supplementary Material]

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

Table 1 below summarizes the results on the quantity $\ell_f(h)$ and $S_f$ for different properties. Note that for a given property, $\ell_f(h)$ is unique while $S_f$ is not.

Table 1: Values of $\ell_f(h)$ and $S_f$ for different properties

| Property | $f_x(p_x)$ | $\ell_f(h)$ | $S_f$ |
|---|---|---|---|
| KL divergence | $p_x \log \frac{p_x}{q_x}$ | $-\min_{x \in \mathcal{X}} \log(hq_x)$ | $\log 2$ |
| $L_1$ distance | $|p_x - q_x| - q_x$ | 1 | 1 |
| Shannon entropy | $p_x \log \frac{1}{p_x}$ | $-\log(h)$ | $\log 2$ |
| Power sum $(a)$ | $p_x^a$ $(a \geq 1)$ | 1 | $a$ |
| Normalized support coverage | $\frac{1 - e^{-mp_x}}{m}$ | 1 | 1 |
| Distance to uniformity | $\left|p_x - \frac{1}{k}\right| - \frac{1}{k}$ | 1 | 1 |

For simplicity, we denote the *partial expectation* $\mathbb{E}_Y[X] \overset{\text{def}}{=} \mathbb{E}[X \mathbb{1}_Y]$, and $a \wedge b \overset{\text{def}}{=} \min\{a, b\}$. To simplify our proofs and expressions, we assume that the number of samples $n \geq 150$, the amplification parameter $t > 2.5$, and $0 < \epsilon \leq 0.1$. Without loss of generality, we also assume that $s_0$, $u_{\max}$ and $r$ are integers. Finally, set $t = c_1 \log^{1/2-\epsilon} n + 1$ and $s_0 = c_2 \log^{2\epsilon} n$, where $c_1$ and $c_2$ are fixed constants such that $1 \geq c_1, c_2 > 0$ and $c_1 \sqrt{c_2} \leq 1/11$.

## 3 Outline

The rest of the supplemental material is organized as follows.

In Section 4.1, we present a few concentration inequalities for Poisson and Binomial random variables that will be used in subsequent proofs. In Section 4.2, we analyze the performance of the modified empirical estimator $f^{ME}$ that estimates $p_x$ by $N_x/n$ instead of $N_x/N$. We show that $f^{ME}$ performs nearly as well as the original empirical estimator $f^E$, but is significantly easier to analyze.

In Section 5, we partition the loss of our estimator, $L_{f^*}(p, nt)$, into three parts: $\mathbb{E}[A^2]$, $\mathbb{E}[B^2]$, and $\mathbb{E}[C^2]$, corresponding to a quantity which is roughly $L_{f^E}(p, nt)$, the loss incurred by $f_L^*$, and the loss incurred by $f_S^*$, respectively.

In Section 6, we bound $\mathbb{E}[A^2]$ by roughly $L_{f^E}(p, nt)$. In Section 7, we bound $\mathbb{E}[B^2]$: in Section 7.1 and 7.2 , we bound the squared bias and variance of $f_L^*$ respectively.

In Section 8.1, we partition the series to be estimated in $\mathbb{E}[C^2]$ into $R_f$ and $K_f$, and show that it suffices to estimate the quantity $K_f$. In Section 8.2, we outline how we construct the linear estimator $f_S^*$ based on $K_f$. Then, we bound term $\mathbb{E}[C^2]$: in Section 8.3 and 8.4, we bound the variance and squared bias of $f_S^*$ respectively. In Section 8.5, we derive a tight bound on $\mathbb{E}[C^2]$.

In Section 9, we prove Theorem 1 based on our previous results.

In Section 10, we demonstrate the practical advantages of our methods through experiments on different properties and distributions. We show that our estimator can even match the performance of the $n \log n$-sample empirical estimator in estimating various properties.

## 4 Preliminary Results

### 4.1 Concentration Inequalities for Poisson and Binomial

The following lemma gives tight tail probability bounds for Poisson and Binomial random variables.

**Lemma 1.** *[24] Let $X$ be a Poisson or Binomial random variable with mean $\mu$, then for any $\delta > 0$,*

$$\mathbb{P}(X \geq (1 + \delta)\mu) \leq \left( \frac{e^\delta}{(1 + \delta)^{(1+\delta)}} \right)^\mu \leq e^{-(\delta^2 \wedge \delta)\mu/3}$$

*and for any $\delta \in (0, 1)$,*

$$\mathbb{P}(X \leq (1 - \delta)\mu) \leq \left( \frac{e^{-\delta}}{(1 - \delta)^{(1-\delta)}} \right)^\mu \leq e^{-\delta^2 \mu/2}.$$

We have the following corollary by choosing different values of $\delta$.

**Lemma 2.** *Let $X$ be a Poisson or Binomial random variable with mean $\mu$,*

$$\mathbb{P}(X \leq \frac{1}{2}\mu) \leq e^{-0.15\mu}, \quad \mathbb{P}(X \leq \frac{1}{3}\mu) \leq e^{-0.30\mu},$$

$$\mathbb{P}(X \leq \frac{1}{5}\mu) \leq e^{-0.478\mu}, \text{ and } \mathbb{P}(X \leq \frac{1}{16}\mu) \leq e^{-0.76\mu}.$$

**Lemma 3.** *Let $N \sim \text{Poi}(n)$,*

$$\mathbb{E}\left[ \sqrt{\frac{n}{N}} \, \middle| \, N \geq 1 \right] \leq 1 + \frac{3}{n}.$$

*Proof.* For $N \geq 1$,

$$\frac{n}{N} \leq \frac{n}{N+1} + \frac{3n}{(N+1)(N+2)},$$

hence,

$$\mathbb{E}\left[ \frac{n}{N} \, \middle| \, N \geq 1 \right] \leq \mathbb{E}\left[ \frac{n}{N+1} \, \middle| \, N \geq 1 \right] + \mathbb{E}\left[ \frac{3n}{(N+1)(N+2)} \, \middle| \, N \geq 1 \right]$$

$$\leq \mathbb{E}\left[ \frac{n}{N+1} \right] + \mathbb{E}\left[ \frac{3n}{(N+1)(N+2)} \right]$$

$$= \mathbb{P}[N \geq 1] + \frac{3}{n}\mathbb{P}[N \geq 2]$$

$$\leq 1 + \frac{3}{n},$$

where the second inequality follows from the fact that $\frac{1}{N+1}$ and $\frac{3n}{(N+1)(N+2)}$ decrease with $N$ and the equality follows as $N \sim \text{Poi}(n)$. $\square$

## 4.2 The Modified Empirical Estimator

The modified empirical estimator

$$f^{ME}(X^N) = \sum_{x \in \mathcal{X}} f_x \left( \frac{N_x}{n} \right)$$

estimates the probability of a symbol not by the fraction $N_x/N$ of times it appeared, but by $N_x/n$, where $n$ is the parameter of the Poisson sampling distribution.

We show that the original and modified empirical estimators have very similar performance.

**Lemma 4.** *For all $n \geq 1$,*

$$\mathbb{E}\left[ \left( f^E(X^N) - f^{ME}(X^N) \right)^2 \right] \leq \frac{\ell_f^2 (1/n)}{n}.$$

*Proof.* By the definition of $\ell_f(h)$, if $N_x \geq 1$,

$$\left| f_x \left( \frac{N_x}{n} \right) - f_x \left( \frac{N_x}{N} \right) \right| \leq \ell_f \left( \frac{1}{n} \right) \left| \frac{N_x}{n} - \frac{N_x}{N} \right| = \ell_f \left( \frac{1}{n} \right) \frac{N_x}{N} \frac{|N - n|}{n},$$

and if $N_x = 0$,

$$\left| f_x \left( \frac{N_x}{n} \right) - f_x \left( \frac{N_x}{N} \right) \right| = 0 \leq \ell_f \left( \frac{1}{n} \right) \frac{N_x}{N} \frac{|N - n|}{n}.$$

Therefore,

$$\mathbb{E}\left[ \left( \sum_{x \in \mathcal{X}} f_x \left( \frac{N_x}{n} \right) - f_x \left( \frac{N_x}{N} \right) \right)^2 \right] \leq \mathbb{E}\left[ \left( \sum_{x \in \mathcal{X}} \ell_f \left( \frac{1}{n} \right) \frac{N_x}{N} \frac{|N - n|}{n} \right)^2 \right]$$

$$\leq \mathbb{E}\left[ \left( \ell_f \left( \frac{1}{n} \right) \frac{|N - n|}{n} \right)^2 \right]$$

$$= \frac{\ell_f^2 (1/n)}{n^2} \mathbb{E}\left[ (N - n)^2 \right]$$

$$= \frac{\ell_f^2 (1/n)}{n},$$

where the last step follows as $N \sim \mathrm{Poi}(n)$ and $\mathbb{E}\left[ (N - n)^2 \right] = \mathrm{Var}[N] = n$. □

## 5 Large and Small Probabilities

Recall that $f^*$ has the following form

$$f^*(X^N, X^{N'}) = f_S^*(X^N, X^{N'}) + f_L^*(X^N, X^{N'}).$$

We can rewrite the property as follows

$$f(p) = f(p) - \mathbb{E}[f^{ME}(X^{N''})] + \mathbb{E}[f_S^{ME}(X^{N''}, X^{N'})] + \mathbb{E}[f_L^{ME}(X^{N''}, X^{N'})].$$

The difference between $f^*(X^N, X^{N'})$ and the actual value $f(p)$ can be partitioned into three terms

$$f^*(X^N, X^{N'}) - f(p) = A + B + C,$$

where

$$A \overset{\text{def}}{=} \mathbb{E}[f^{ME}(X^{N''}) - f(p)]$$

is the bias of the modified empirical estimator with $\mathrm{Poi}(nt)$ samples,

$$B \overset{\text{def}}{=} f_L^*(X^N, X^{N'}) - \mathbb{E}[f_L^{ME}(X^{N''}, X^{N'})]$$

corresponds to the loss incurred by the large-probability estimator $f_L^*$, and

$$C \stackrel{\text{def}}{=} f_S^*(X^N, X^{N'}) - \mathbb{E}[f_S^{ME}(X^{N''}, X^{N'})]$$

corresponds to the loss incurred by the small-probability estimator $f_S^*$.

By Cauchy-Schwarz inequality, upper bounds on $\mathbb{E}[A^2]$, $\mathbb{E}[B^2]$, and $\mathbb{E}[C^2]$, suffice to also upper bound the estimation loss $L_{f^*}(p, 2n) = \mathbb{E}[(f^*(X^N, X^{N'}) - f(p))^2]$.

In the next section, we bound the squared bias term $\mathbb{E}[A^2]$. In Section 6 and Section 7, we bound the large- and small-probability terms $\mathbb{E}[B^2]$ and $\mathbb{E}[C^2]$, respectively.

# 6 Squared Bias: $\mathbb{E}[A^2]$

We relate $\mathbb{E}[A^2]$ to $L_{f^E}(p, nt)$ through the following inequality.

**Lemma 5.** *Let $T$ be a positive function over $\mathbb{N}$,*

$$\mathbb{E}[A^2] \leq \frac{1 + T(n)}{nt} \ell_f^2 \left( \frac{1}{nt} \right) + \left( 1 + \frac{1}{T(n)} \right) L_{f^E}(p, nt).$$

*Proof.* We upper bound $\mathbb{E}[A^2]$ in terms of $L_{f^E}(p, nt)$ using Cauchy-Schwarz inequality and Lemma 4.

$$\mathbb{E}[A^2] = \left( \sum_{x \in \mathcal{X}} \left( \mathbb{E}\left[ f_x \left( \frac{N_x''}{nt} \right) \right] - f_x(p_x) \right) \right)^2$$

$$= \left( \sum_{x \in \mathcal{X}} \left( \mathbb{E}\left[ f_x \left( \frac{N_x''}{nt} \right) \right] - \mathbb{E}\left[ f_x \left( \frac{N_x''}{N''} \right) \right] \right) + \sum_{x \in \mathcal{X}} \left( \mathbb{E}\left[ f_x \left( \frac{N_x''}{N''} \right) \right] - f_x(p_x) \right) \right)^2$$

$$\leq (1 + T(n)) \left( \sum_{x \in \mathcal{X}} \left( \mathbb{E}\left[ f_x \left( \frac{N_x''}{nt} \right) \right] - \mathbb{E}\left[ f_x \left( \frac{N_x''}{N''} \right) \right] \right) \right)^2 + \left( 1 + \frac{1}{T(n)} \right) L_{f^E}(p, nt)$$

$$\leq \frac{1 + T(n)}{nt} \ell_f^2 \left( \frac{1}{nt} \right) + \left( 1 + \frac{1}{T(n)} \right) L_{f^E}(p, nt).$$

$\square$

# 7 Large Probabilities: $\mathbb{E}[B^2]$

Note that

$$\mathbb{E}[B^2] = \mathbb{E}[(f_L^*(X^N, X^{N'}) - \mathbb{E}[f_L^{ME}(X^{N''}, X^{N'})])^2]$$
$$= Bias(f_L^*)^2 + Var(f_L^*),$$

where

$$\text{Bias}(f_L^*) \stackrel{\text{def}}{=} \mathbb{E}[f_L^*(X^N, X^{N'}) - f_L^{ME}(X^{N''}, X^{N'})]$$

and

$$\text{Var}(f_L^*) \stackrel{\text{def}}{=} \mathbb{E}[(f_L^*(X^N, X^{N'}) - \mathbb{E}[f_L^*(X^N, X^{N'})])^2]$$

are the bias and variance of $f_L^*(X^N, X^{N'})$ in estimating $\mathbb{E}[f_L^{ME}(X^{N''}, X^{N'})]$, respectively. We shall upper bound the absolute bias and variance as

$$|\text{Bias}(f_L^*)| \leq \sqrt{(8S_f)^2 \left( \frac{1}{s_0} \wedge \frac{k}{n} \right) + 6\ell_f^2 \left( \frac{1}{nt} \right) \frac{1}{n}}$$

and

$$\text{Var}(f_L^*) \leq \ell_f^2 \left( \frac{1}{n} \right) \frac{4s_0}{n}$$

in Section 7.1 and Section 7.2 respectively. It follows that

**Lemma 6.** *For $t > 2.5$ and $s_0 \geq 1$,*

$$\mathbb{E}[B^2] = Bias(f_L^*)^2 + Var(f_L^*) \leq (8S_f)^2 \left( \frac{1}{s_0} \wedge \frac{k}{n} \right) + 10\ell_f^2 \left( \frac{1}{nt} \right) \frac{s_0}{n}.$$

## 7.1 Bounding the Bias of $f_L^*$

To bound the bias of $f_L^*$, we need the following lemma.

**Lemma 7.** *[25] For any binomial random variable $X \sim B(n, p)$, continuous function $f_0$, and $p \in [0, 1]$,*

$$\left| \mathbb{E}\left[ f_0\left( \frac{X}{n} \right) \right] - f_0(p) \right| \leq 3\omega_{f_0}^2 \left( \sqrt{\frac{p(1-p)}{n}} \right).$$

Recall that $\omega_f^2(h) \leq S_f h$ from our assumption.

**Lemma 8.** *For $n \geq 150$,*

$$\left| \mathbb{E}_{N \geq 1}\left[ f_x\left( \frac{N_x}{n} \right) - f_x(p_x) \right] \right| \leq \ell_f\left( \frac{1}{n} \right) \frac{p_x}{\sqrt{n}} + 3.06 S_f \sqrt{\frac{p_x}{n}}.$$

*Proof.* Noting $n \geq 150$, it follows from Lemma 3 and Lemma 6 that

$$\left| \mathbb{E}_{N \geq 1}\left[ f_x\left( \frac{N_x}{n} \right) - f_x(p_x) \right] \right| \leq \left| \mathbb{E}_{N \geq 1}\left[ f_x\left( \frac{N_x}{n} \right) - f_x\left( \frac{N_x}{N} \right) \right] \right| + \left| \mathbb{E}_{N \geq 1}\left[ f_x\left( \frac{N_x}{N} \right) - f_x(p_x) \right] \right|$$

$$\leq \ell_f\left( \frac{1}{n} \right) \frac{p_x}{n} \mathbb{E}[|N - n|] + \mathbb{E}\left[ 3\omega_f^2\left( \sqrt{\frac{p_x(1-p_x)}{N}} \right) \middle| N \geq 1 \right]$$

$$\leq \ell_f\left( \frac{1}{n} \right) \frac{p_x}{n} \sqrt{\mathbb{E}[(N-n)^2]} + 3S_f \sqrt{\frac{p_x}{n}} \mathbb{E}\left[ \sqrt{\frac{n}{N}} \middle| N \geq 1 \right]$$

$$\leq \ell_f\left( \frac{1}{n} \right) \frac{p_x}{\sqrt{n}} + 3.06 S_f \sqrt{\frac{p_x}{n}}.$$

$\square$

The next lemma essentially bounds the individual bias term for each symbol $x$.

**Lemma 9.** *For $t > 2.5$,*

$$\left| \mathbb{E}\left[ f_x\left( \frac{N_x}{n} \right) - f_x\left( \frac{N_x''}{nt} \right) \right] \right| \leq 5 S_f \sqrt{\frac{p_x}{n}} + 1.65 \ell_f\left( \frac{1}{nt} \right) \frac{p_x}{\sqrt{n}}.$$

*Proof.* Using Lemma 8,

$$\left| \mathbb{E}\left[ f_x\left( \frac{N_x}{n} \right) - f_x\left( \frac{N_x''}{nt} \right) \right] \right|$$

$$\leq \left| \mathbb{E}_{N, N'' \geq 1}\left[ f_x\left( \frac{N_x}{n} \right) - f_x\left( \frac{N_x''}{nt} \right) \right] \right| + \ell_f\left( \frac{1}{n} \right) \mathbb{E}\left[ \frac{N_x}{n} \right] e^{-n} + \ell_f\left( \frac{1}{nt} \right) \mathbb{E}\left[ \frac{N_x''}{nt} \right] e^{-nt}$$

$$\leq \left| \mathbb{E}_{N \geq 1}\left[ f_x\left( \frac{N_x}{n} \right) - f_x(p_x) \right] \right| + \left| \mathbb{E}_{N'' \geq 1}\left[ f_x(p_x) - f_x\left( \frac{N_x''}{nt} \right) \right] \right| + 2\ell_f\left( \frac{1}{nt} \right) p_x e^{-n}$$

$$\leq 5 S_f \sqrt{\frac{p_x}{n}} + 1.65 \ell_f\left( \frac{1}{nt} \right) \frac{p_x}{\sqrt{n}},$$

where the last step follows from $\ell_f\left( \frac{1}{n} \right) \leq \ell_f\left( \frac{1}{nt} \right)$, $e^{-n} \leq \sqrt{n}$, and $t > 2.5$. $\square$

Finally, the next lemma bounds the absolute bias of $f_L^*$.

**Lemma 10.** *For $t > 2.5$ and $s_0 \geq 1$,*

$$|Bias(f_L^*)| \leq \sqrt{(8 S_f)^2 \left( \frac{1}{s_0} \wedge \frac{k}{n} \right) + 6\ell_f^2\left( \frac{1}{nt} \right) \frac{1}{n}}.$$

*Proof.*

$$|Bias(f_L^*)| = \left| \mathbb{E}\left[ \sum_{x \in \mathcal{X}} f_x\left(\frac{N_x}{n}\right) \mathbb{1}_{N_x' > s_0} - \sum_{x \in \mathcal{X}} \mathbb{E}[\mathbb{1}_{N_x > s_0}] \mathbb{E}\left[ f_x\left(\frac{N_x''}{nt}\right) \right] \right] \right|$$

$$\overset{(a)}{\leq} \sum_{x \in \mathcal{X}} \mathbb{E}[\mathbb{1}_{N_x > s_0}] \left| \mathbb{E}\left[ f_x\left(\frac{N_x}{n}\right) - f_x\left(\frac{N_x''}{nt}\right) \right] \right|$$

$$\overset{(b)}{\leq} \sum_{x \in \mathcal{X}} \mathbb{E}[\mathbb{1}_{N_x > s_0}] \left( 5S_f \sqrt{\frac{p_x}{n}} + 1.65\ell_f\left(\frac{1}{nt}\right)\frac{p_x}{\sqrt{n}} \right)$$

$$\overset{(c)}{\leq} \sqrt{\frac{1}{n}} 5S_f \sum_{x \in \mathcal{X}} \mathbb{E}[\mathbb{1}_{N_x > s_0}]\sqrt{p_x} + 1.65\ell_f\left(\frac{1}{nt}\right)\frac{1}{\sqrt{n}}$$

$$\overset{(d)}{\leq} \sqrt{\frac{1}{n}} 5S_f \sqrt{\left(\sum_{x \in \mathcal{X}} \mathbb{E}[\mathbb{1}_{N_x > s_0}]\right)\left(\sum_{x \in \mathcal{X}} \mathbb{E}[\mathbb{1}_{N_x > s_0}]p_x\right)} + 1.65\ell_f\left(\frac{1}{nt}\right)\frac{1}{\sqrt{n}}$$

$$\overset{(e)}{\leq} 5S_f \sqrt{\frac{1}{s_0} \wedge \frac{k}{n}} + 1.65\ell_f\left(\frac{1}{nt}\right)\frac{1}{\sqrt{n}}$$

$$\overset{(f)}{\leq} \sqrt{(8S_f)^2\left(\frac{1}{s_0} \wedge \frac{k}{n}\right) + 6\ell_f^2\left(\frac{1}{nt}\right)\frac{1}{n}},$$

where $(a)$ follows from triangle inequality, $(b)$ follows from Lemma 9, $(c)$ follows as $\sum_{x \in \mathcal{X}} p_x = 1$ and $\mathbb{E}[\mathbb{1}_{N_x > s_0}] \leq 1$, $(d)$ follows from Cauchy-Schwarz inequality, $(e)$ follows from Markov inequality, i.e., $\mathbb{E}[\mathbb{1}_{N_x > s_0}] = \mathbb{P}[N_x > s_0] \leq np_x/s_0$ and $\sum_{x \in \mathcal{X}} \mathbb{E}[\mathbb{1}_{N_x > s_0}] \leq k$, and $(f)$ follows from the inequality $a + b \leq \sqrt{2(a^2 + b^2)}$. $\qquad\square$

### 7.2 Bounding the Variance of $f_L^*$

The following lemma exploits independence and bounds the variance of $f_L^*$.

**Lemma 11.** *For $s_0 \geq 1$,*

$$\mathrm{Var}\,(f_L^*) \leq \ell_f^2\left(\frac{1}{n}\right)\frac{4s_0}{n}.$$

*Proof.* Due to independence,

$$\mathrm{Var}\,(f_L^*) = \mathrm{Var}\left( \sum_{x \in \mathcal{X}} f_x\left(\frac{N_x}{n}\right) \mathbb{1}_{N_x' > s_0} \right)$$

$$= \sum_{x \in \mathcal{X}} \mathrm{Var}\left( f_x\left(\frac{N_x}{n}\right) \mathbb{1}_{N_x' > s_0} \right)$$

$$= \sum_{x \in \mathcal{X}} \mathrm{Var}(\mathbb{1}_{N_x' > s_0})\mathbb{E}\left[ f_x^2\left(\frac{N_x}{n}\right) \right] + \sum_{x \in \mathcal{X}} \left( \mathbb{E}[\mathbb{1}_{N_x' > s_0}] \right)^2 \mathrm{Var}\left( f_x\left(\frac{N_x}{n}\right) \right)$$

$$\leq \sum_{x \in \mathcal{X}} Var(\mathbb{1}_{N_x' > s_0})\mathbb{E}\left[ f_x^2\left(\frac{N_x}{n}\right) \right] + \sum_{x \in \mathcal{X}} \mathrm{Var}\left( f_x\left(\frac{N_x}{n}\right) \right).$$

To bound the first term,

$$\mathrm{Var}(\mathbb{1}_{N_x' > s_0})\mathbb{E}\left[ f_x^2\left(\frac{N_x}{n}\right) \right] \leq \mathrm{Var}(\mathbb{1}_{N_x' > s_0})\mathbb{E}\left[ \ell_f^2\left(\frac{1}{n}\right)\left(\frac{N_x}{n}\right)^2 \right]$$

$$\leq \ell_f^2\left(\frac{1}{n}\right)\frac{p_x}{n}\left( 1 + np_x\mathrm{Var}(\mathbb{1}_{N_x' > s_0}) \right),$$

where Lemma 2 further bounds the final term by

$$\text{Var}(\mathbb{1}_{N'_x > s_0})p_x \le \mathbb{P}[N'_x \le s_0]p_x$$

$$= e^{-np_x} \sum_{i=0}^{s_0} \frac{(np_x)^{i+1}}{(i+1)!} \frac{i+1}{n}$$

$$\le \frac{s_0+1}{n} e^{-np_x} \sum_{i=0}^{s_0} \frac{(np_x)^{i+1}}{(i+1)!}$$

$$= \frac{s_0+1}{n} \mathbb{P}(1 \le N'_x \le s_0+1)$$

$$\le \frac{s_0+1}{n}.$$

To bound the second term, let $\hat{N}_x$ be an i.i.d. copy of $N_x$ for each $x$,

$$2\text{Var}\left(f_x\left(\frac{N_x}{n}\right)\right) = \text{Var}\left(f_x\left(\frac{N_x}{n}\right) - f_x\left(\frac{\hat{N}_x}{n}\right)\right)$$

$$= \mathbb{E}\left[\left(f_x\left(\frac{N_x}{n}\right) - f_x\left(\frac{\hat{N}_x}{n}\right)\right)^2\right]$$

$$\le \mathbb{E}\left[\ell_f^2\left(\frac{1}{n}\right)\left(\frac{N_x}{n} - \frac{\hat{N}_x}{n}\right)^2\right]$$

$$= 2\ell_f^2\left(\frac{1}{n}\right)\frac{p_x}{n}.$$

A simple combination of these bounds yields the lemma. $\qquad\square$

# 8 Small Probabilities: $\mathbb{E}[C^2]$

As outlined in Section 1, the quantity to be estimated in $C$ is

$$\mathbb{E}[f_S^{ME}(X^{N''}, X^{N'})] = \sum_{x \in \mathcal{X}} \mathbb{E}[\mathbb{1}_{N_x \le s_0}]\mathbb{E}\left[f_x\left(\frac{N''_x}{nt}\right)\right] = \sum_{x \in \mathcal{X}} \mathbb{E}[\mathbb{1}_{N_x \le s_0}] \sum_{v=1}^{\infty} e^{-\lambda_x t}\frac{(\lambda_x t)^v}{v!} f_x\left(\frac{v}{nt}\right).$$

We truncate the inner summation according to the threshold $u_{\max} = 2s_0 t + 2s_0 - 1$ and define

$$K_f \overset{\text{def}}{=} \sum_{x \in \mathcal{X}} \mathbb{E}[\mathbb{1}_{N_x \le s_0}] \sum_{v=1}^{u_{\max}} e^{-\lambda_x t}\frac{(\lambda_x t)^v}{v!} f_x\left(\frac{v}{nt}\right)$$

and

$$R_f \overset{\text{def}}{=} \sum_{x \in \mathcal{X}} \mathbb{E}[\mathbb{1}_{N_x \le s_0}] \sum_{v=u_{\max}+1}^{\infty} e^{-\lambda_x t}\frac{(\lambda_x t)^v}{v!} f_x\left(\frac{v}{nt}\right),$$

then,

$$\mathbb{E}[f_S^{ME}(X^{N''}, X^{N'})] = K_f + R_f.$$

The truncation threshold $u_{\max}$ is calibrated such that for each symbol $x$,

$$\sum_{v=1}^{u_{\max}} e^{-\lambda_x t}\frac{(\lambda_x t)^v}{v!} f_x\left(\frac{v}{nt}\right)$$

contains only roughly $\log(n)$ terms and $R_f^2$ is sufficiently small and contributes only to the slack term in Theorem 1, as shown in Lemma 13. In Section 8.2, we shall thus construct $f_S^*(X^N, X^{N'})$ to estimate $K_f$ instead of $\mathbb{E}[f_S^{ME}(X^{N''}, X^{N'})]$.

Analogous to Section 7, define

$$\text{Bias}(f_S^*) \overset{\text{def}}{=} \mathbb{E}[f_S^*(X^N, X^{N'}) - K_f]$$

and

$$\text{Var}(f_S^*) \overset{\text{def}}{=} \mathbb{E}[(f_S^*(X^N, X^{N'}) - \mathbb{E}[f_S^*(X^N, X^{N'})])^2]$$

as the bias and variance of $f_S^*(X^N, X^{N'})$ in estimating $K_f$, respectively, it follows that

$$
\begin{aligned}
\mathbb{E}[C^2] &= \mathbb{E}[(f_S^*(X^N, X^{N'}) - \mathbb{E}[f_L^{ME}(X^{N''}, X^{N'})])^2] \\
&= \mathbb{E}\left[\left(f_S^*(X^N, X^{N'}) - (K_f + R_f)\right)^2\right] \\
&= \text{Var}(f_S^*) + (\text{Bias}(f_S^*) - R_f)^2 \\
&\leq \text{Var}(f_S^*) + (1 + \log n)(\text{Bias}(f_S^*))^2 + \left(1 + \frac{1}{\log n}\right) R_f^2.
\end{aligned}
$$

We shall upper bound the variance and squared bias as

$$\text{Var}(f_S^*) \leq (n \wedge k)\left(\ell_f\left(\frac{1}{nt}\right)\frac{u_{\max}}{nt}\right)^2 e^{4r(t-1)}.$$

and

$$\text{Bias}(f_S^*)^2 \leq \left(1 \wedge \frac{k^2}{n^2}\right)e^{-4s_0 t}\ell_f^2\left(\frac{1}{nt}\right)$$

in Section 8.3 and Section 8.4 respectively. It follows by simple algebraic manipulation that

**Lemma 12.** *For the set of parameters specified in Section 2, if $c_1\sqrt{c_2} \leq 1/11$, $t > 2.5$, $n \geq 150$, and $1 \leq s_0 \leq \log^{0.2}(n)$,*

$$\mathbb{E}[C^2] \leq 13^2\left(1 \wedge \frac{k}{n}\right)\ell_f^2\left(\frac{1}{nt}\right)\left(\frac{\log^2 n}{e^{0.6 s_0}}\right).$$

## 8.1 Bounding the Last Few Terms

We now show that $R_f^2$ is sufficiently small and only contributes to the slack term in Theorem 1. The key is to divide the sum into two parts and apply Lemma 2 seperately.

**Lemma 13.** *For $n \geq 150$, $1 \leq s_0 \leq \log^{0.2} n$, and $t > 2.5$,*

$$R_f^2 \leq \left(7.1\left(1 \wedge \frac{k}{n}\right)\ell_f\left(\frac{1}{n}\right)e^{-0.3 s_0}\log(n)\right)^2 + \left(\frac{7.1}{n^{3.8}}\ell_f\left(\frac{1}{n}\right)\right)^2.$$

*Proof.* Recall that $u_{\max} = 2s_0 t + 2s_0$, we upper bound the absolute value of $R_f$ as

$$
\begin{aligned}
|R_f| &= \left|\sum_{x \in \mathcal{X}} \sum_{u=0}^{s_0} \sum_{v=2s_0 t + 2s_0}^{\infty} e^{-\lambda_x} \frac{\lambda_x^u}{u!} e^{-\lambda_x t} \frac{(\lambda_x t)^v}{v!} f_x\left(\frac{v}{nt}\right)\right| \\
&\leq \sum_{x \in \mathcal{X}} \sum_{u+v=2s_0 t + 2s_0}^{\infty} e^{-\lambda_x(t+1)} \frac{(\lambda_x(t+1))^{u+v}}{(u+v)!} \cdot \\
&\quad \left(\ell_f\left(\frac{2s_0 t + 2s_0}{nt}\right)\frac{u+v}{nt}\right) \sum_{u=0}^{s_0}\binom{u+v}{u}\left(\frac{1}{t+1}\right)^u\left(\frac{t}{t+1}\right)^v \\
&= \sum_{x \in \mathcal{X}} \sum_{u+v=2s_0 t + 2s_0}^{\infty} e^{-\lambda_x(t+1)} \frac{(\lambda_x(t+1))^{u+v}}{(u+v)!} \cdot \\
&\quad \left(\ell_f\left(\frac{2s_0 t + 2s_0}{nt}\right)\frac{u+v}{nt}\right)\mathbb{P}\left(B\left(u+v, \frac{1}{t+1}\right) \leq s_0\right) \\
&\leq \ell_f\left(\frac{1}{n}\right) \sum_{x \in \mathcal{X}} \sum_{u+v=2s_0 t + 2s_0}^{\infty} e^{-\lambda_x(t+1)} \frac{(\lambda_x(t+1))^{u+v}}{(u+v)!}\frac{u+v}{nt}\mathbb{P}\left(B\left(u+v, \frac{1}{t+1}\right) \leq s_0\right).
\end{aligned}
$$

For $u + v \geq 2s_0 t + 2s_0$, Lemma 2 yields

$$\mathbb{P}\left(B\left(u+v, \frac{1}{t+1}\right) \leq s_0\right) \leq e^{-0.15(u+v)/(t+1)} \leq e^{-0.3s_0}.$$

Truncate the inner summation at $u + v = 5(t+1)\log n$ and apply the above inequality,

$$\sum_{x \in \mathcal{X}} \sum_{u+v=2s_0 t + 2s_0}^{5(t+1)\log n} e^{-\lambda_x(t+1)} \frac{(\lambda_x(t+1))^{u+v}}{(u+v)!} \frac{u+v}{nt} \mathbb{P}\left(B\left(u+v, \frac{1}{t+1}\right) \leq s_0\right)$$

$$\leq \frac{5(t+1)\log n}{nt} e^{-0.3s_0} \sum_{x \in \mathcal{X}} \sum_{u+v=2s_0 t + 2s_0}^{5(t+1)\log n} e^{-\lambda_x(t+1)} \frac{(\lambda_x(t+1))^{u+v}}{(u+v)!}$$

$$\leq \frac{5(t+1)\log n}{nt} e^{-0.3s_0} \sum_{x \in \mathcal{X}} \mathbb{P}\left(\mathrm{Poi}(\lambda_x(t+1)) \geq 2s_0 t + 2s_0\right)$$

$$\leq \frac{5(t+1)\log n}{nt} e^{-0.3s_0} \sum_{x \in \mathcal{X}} (1 \wedge \lambda_x)$$

$$\leq 7\left(1 \wedge \frac{k}{n}\right) e^{-0.3s_0} \log n,$$

where the second last inequality follows from the Markov's inequality and the last one follows from $\sum_{x \in \mathcal{X}} \lambda_x = n$ and $|\mathcal{X}| = k$.

For $u + v \geq 5(t+1)\log n + 1$, Lemma 2, $1 \leq s_0 \leq \log^{0.2} n$, and $n \geq 150$ together yield

$$\frac{u+v}{t+1} \geq 5\log n \geq 16 \log^{0.2} n \geq 16 s_0$$

and

$$\mathbb{P}\left(B\left(u+v, \frac{1}{t+1}\right) \leq s_0\right) \leq e^{-0.76 \times 5 \log n} \leq \frac{1}{n^{3.8}}.$$

It remains to consider the following partial sum.

$$\sum_{x \in \mathcal{X}} \sum_{u+v=5(t+1)\log n + 1}^{\infty} e^{-\lambda_x(t+1)} \frac{(\lambda_x(t+1))^{u+v}}{(u+v)!} \frac{u+v}{nt} \mathbb{P}\left(B\left(u+v, \frac{1}{t+1}\right) \leq s_0\right)$$

$$\leq \frac{1}{n^{3.8}} \frac{1}{nt} \sum_{x \in \mathcal{X}} \sum_{u+v=5(t+1)\log n + 1}^{\infty} e^{-\lambda_x(t+1)} \frac{(\lambda_x(t+1))^{u+v}}{(u+v)!} (u+v)$$

$$\leq \frac{1}{n^{3.8}} \frac{1}{nt} \sum_{x \in \mathcal{X}} \lambda_x(t+1)$$

$$\leq \frac{1.4}{n^{3.8}},$$

where the last inequality comes from $\sum_{x \in \mathcal{X}} \lambda_x = n$ and $t > 2.5$. The lemma follows from Cauchy-Schwarz inequality. $\qquad \square$

## 8.2 Estimator Construction for Small Probabilities: $f_S^*$

According to Lemma 13, it suffices to estimate

$$K_f = \sum_{x \in \mathcal{X}} \mathbb{E}[\mathbb{1}_{N_x \leq s_0}] \sum_{u=1}^{u_{\max}} e^{-\lambda_x t} \frac{(\lambda_x t)^u}{u!} f_x\left(\frac{u}{nt}\right).$$

Recall that

$$g_x(u) = f_x\left(\frac{u}{nt}\right)\left(\frac{t}{t-1}\right)^u,$$

we can rewrite $K_f$ as

$$K_f = \sum_{x \in \mathcal{X}} \mathbb{E}[\mathbb{1}_{N_x \leq s_0}] e^{-\lambda_x} \sum_{u=1}^{u_{\max}} e^{-\lambda_x(t-1)} \frac{(\lambda_x(t-1))^u}{u!} g_x(u).$$

Let

$$f_u(y) \overset{\text{def}}{=} J_{2u}(2\sqrt{y}) = \sum_{i=0}^{\infty} \frac{(-1)^i y^{i+u}}{i!(i+2u)!},$$

where $J_{2u}$ is the Bessel function of the first kind with parameter $2u$. Our estimator is motivated by the following equality.

**Lemma 14.** *For any $u \in \mathbb{Z}^+$ and $y \geq 0$,*

$$\int_0^{\infty} e^{-\alpha} \alpha^u f_u(\alpha y) d\alpha = e^{-y} y^u.$$

*Proof.* By Fubini's theorem and the series expansion of $f_u$,

$$\int_0^{\infty} e^{-\alpha} \alpha^u f_u(\alpha y) d\alpha = \int_0^{\infty} e^{-\alpha} \alpha^u \sum_{i=0}^{\infty} \frac{(-1)^i (\alpha y)^{i+u}}{(i!)(i+2u)!} d\alpha$$

$$= \sum_{i=0}^{\infty} \frac{(-1)^i (y)^{i+u}}{(i!)(i+2u)!} \int_0^{\infty} e^{-\alpha} \alpha^{i+2u} d\alpha.$$

Observe that the integral is actually $\Gamma(i + 2u + 1)$ and equals to $(i + 2u)!$,

$$\sum_{i=0}^{\infty} \frac{(-1)^i (y)^{i+u}}{(i!)(i+2u)!} \int_0^{\infty} e^{-\alpha} \alpha^{i+2u} d\alpha = \sum_{i=0}^{\infty} \frac{(-1)^i (y)^{i+u}}{(i!)(i+2u)!} (i+2u)!$$

$$= \sum_{i=0}^{\infty} \frac{(-1)^i (y)^{i+u}}{i!}$$

$$= e^{-y} y^u.$$

$\square$

Therefore, let

$$h_x(\lambda_x) \overset{\text{def}}{=} e^{-\lambda_x} \sum_{u=1}^{u_{\max}} \frac{g_x(u)}{u!} \left( \int_0^{\infty} e^{-\alpha} \alpha^u f_u(\alpha \lambda_x (t-1)) d\alpha \right),$$

we can rewrite

$$K_f = \sum_{x \in \mathcal{X}} \mathbb{E}[\mathbb{1}_{N_x \leq s_0}] h_x(\lambda_x).$$

We apply the *polynomial smoothing technique* in [22] and approximate $h_x(y)$ by

$$\hat{h}_x(\lambda_x) \overset{\text{def}}{=} e^{-\lambda_x} \sum_{u=1}^{u_{\max}} \frac{g_x(u)}{u!} \left( \int_0^{r} e^{-\alpha} \alpha^u f_u(\alpha \lambda_x (t-1)) d\alpha \right),$$

where $r$ is the polynomial smoothing parameter defined in Section 1.

We now expand $\hat{h}_x(\lambda_x)$ as a product of $e^{-\lambda_x}$ and a power series of $\lambda_x$.

**Lemma 15.** *For $t > 2.5$,*

$$\hat{h}_x(\lambda_x) = e^{-\lambda_x} \sum_{v=1}^{\infty} h_{x,v} \lambda_x^v,$$

*where*

$$h_{x,v} = (t-1)^v \sum_{u=1}^{(u_{\max} \wedge v)} \frac{g_x(u)(-1)^{v-u}}{(v-u)!u!} \left( 1 - e^{-r} \sum_{j=0}^{v+u} \frac{r^j}{j!} \right).$$

*Proof.* By Fubini's theorem and the series expansion of $f_u$,

$$\int_0^r e^{-\alpha}\alpha^u f_u(\alpha\lambda_x(t-1))d\alpha = \int_0^r e^{-\alpha}\alpha^u \sum_{i=0}^{\infty} \frac{(-1)^i(\alpha\lambda_x(t-1))^{i+u}}{(i!)(i+2u)!}d\alpha$$

$$= \sum_{i=0}^{\infty} \frac{(-1)^i(\lambda_x(t-1))^{i+u}}{(i!)(i+2u)!}\int_0^r e^{-\alpha}\alpha^{i+2u}d\alpha$$

$$= \sum_{i=0}^{\infty} \frac{(-1)^i(\lambda_x(t-1))^{i+u}}{i!}\left(1 - e^{-r}\sum_{j=0}^{i+2u}\frac{r^j}{j!}\right).$$

Hence,

$$\hat{h}_x(\lambda_x) = e^{-\lambda_x}\sum_{u=1}^{u_{\max}}\frac{g_x(u)}{u!}\left(\int_0^r e^{-\alpha}\alpha^u f_u(\alpha\lambda_x(t-1))d\alpha\right)$$

$$= e^{-\lambda_x}\sum_{u=1}^{u_{\max}}\frac{g_x(u)}{u!}\sum_{i=0}^{\infty}\frac{(-1)^i(\lambda_x(t-1))^{i+u}}{i!}\left(1 - e^{-r}\sum_{j=0}^{i+2u}\frac{r^j}{j!}\right)$$

$$= e^{-\lambda_x}\sum_{v=1}^{\infty}\left[(t-1)^v\sum_{u=1}^{(u_{\max}\wedge v)}\frac{g_x(u)(-1)^{v-u}}{(v-u)!u!}\left(1 - e^{-r}\sum_{j=0}^{v+u}\frac{r^j}{j!}\right)\right]\lambda_x^v$$

$$= e^{-\lambda_x}\sum_{v=1}^{\infty}h_{x,v}\lambda_x^v$$

$\square$

An unbiased estimator of $\hat{h}_x(\lambda_x) = e^{-\lambda_x}\sum_{v=1}^{\infty}h_{x,v}\lambda_x^v$ is

$$\sum_{v=1}^{\infty}h_{x,v}v!\cdot\mathbb{1}_{N_x=v} = h_{x,N_x}\cdot N_x!.$$

Our small-probability estimator is thus

$$f_S^*(X^N, X^{N'}) = \sum_{x\in\mathcal{X}}h_{x,N_x}\cdot N_x!\cdot\mathbb{1}_{N'_x\leq s_0}.$$

In the next section, we show that the connection between $h_x(\lambda)$ and $\hat{h}_x(\lambda)$ leads to a small expected squared loss of $f_S^*$.

## 8.3 Bounding the Variance of $f_S^*$

First we upper bound the variance of $f_S^*$ in terms of the coefficients $h_{x,v}$.

**Lemma 16.** *The variance of $f_S^*$ is bounded by*

$$\mathrm{Var}(f_S^*) \leq (n\wedge k)\max_{x\in\mathcal{X}}\max_v h_{x,v}^2 v!^2.$$

*Proof.* First observe that independence and $\mathrm{Var}[X] \leq \mathbb{E}[X^2]$ imply

$$\mathrm{Var}(f_S^*) = \mathrm{Var}(\sum_{x\in\mathcal{X}}\sum_{v=1}^{\infty}h_{x,v}v!\mathbb{1}_{N_x=v}\mathbb{1}_{N'_x\leq s_0})$$

$$= \sum_{x\in\mathcal{X}}\mathrm{Var}(\sum_{v=1}^{\infty}h_{x,v}v!\mathbb{1}_{N_x=v}\mathbb{1}_{N'_x\leq s_0})$$

$$\leq \sum_{x\in\mathcal{X}}\mathbb{E}[(\sum_{v=1}^{\infty}h_{x,v}v!\mathbb{1}_{N_x=v}\mathbb{1}_{N'_x\leq s_0})^2].$$

Note that $\mathbb{1}_{N_x=u}\mathbb{1}_{N_x=v} = 0$ for any $u \neq v$, we can rewrite the last summation as

$$\sum_{x \in \mathcal{X}} \mathbb{E}[\sum_{v=1}^{\infty} (h_{x,v}v!)^2 \mathbb{1}_{N_x=v} \mathbb{1}_{N'_x \leq s_0}] \leq \max_{x \in \mathcal{X}} \max_{v} h_{x,v}^2 v!^2 \mathbb{E}[\sum_{x \in \mathcal{X}} \sum_{v=1}^{\infty} \mathbb{1}_{N_x=v} \mathbb{1}_{N'_x \leq s_0}]$$

$$\leq \max_{x \in \mathcal{X}} \max_{v} h_{x,v}^2 v!^2 \mathbb{E}[\sum_{x \in \mathcal{X}} \sum_{v=1}^{\infty} \mathbb{1}_{N_x=v}]$$

$$\leq (n \wedge k) \max_{x \in \mathcal{X}} \max_{v} h_{x,v}^2 v!^2,$$

where the last inequality follows from $\sum_{x \in \mathcal{X}} \sum_{v=1}^{\infty} \mathbb{1}_{N_x=v} \leq N \wedge k$ and $\mathbb{E}[N] = n$. $\qquad\square$

The following lemma provides a uniform bound on $|h_{x,v}v!|$, which, by Lemma 16, is sufficient to bound the variance of $f_S^*$.

**Lemma 17.** *For* $t > 2.5$,

$$|h_{x,v}v!| \leq \ell_f\left(\frac{1}{nt}\right)\frac{u_{\max}}{nt}e^{2r(t-1)}.$$

*Proof.* From the definition of $g_x(u)$,

$$|h_{x,v}v!| \leq (t-1)^v e^{-r} \sum_{u=1}^{(u_{\max}\wedge v)} \frac{|g_x(u)|v!}{(v-u)!u!} \sum_{j=v+u+1}^{\infty} \frac{r^j}{j!}$$

$$= e^{-r} \sum_{u=1}^{(u_{\max}\wedge v)} \left|f_x\left(\frac{u}{nt}\right)\right| t^u (t-1)^{v-u} \binom{v}{u} \sum_{j=v+u+1}^{\infty} \frac{r^j}{j!}$$

$$\leq \ell_f\left(\frac{1}{nt}\right)\frac{u_{\max}}{nt}e^{-r} \sum_{u=1}^{(u_{\max}\wedge v)} t^u (t-1)^{v-u} \binom{v}{u} \sum_{j=v+u+1}^{\infty} \frac{r^j}{j!}$$

$$\leq \ell_f\left(\frac{1}{nt}\right)\frac{u_{\max}}{nt}e^{-r} \sum_{j=v+2}^{\infty} \frac{r^j}{j!} \sum_{u=1}^{(u_{\max}\wedge v)} \binom{v}{u} t^u (t-1)^{v-u}.$$

For $t > 2.5$, the binomial expansion theorem yields

$$\sum_{u=1}^{(u_{\max}\wedge v)} \binom{v}{u} t^u (t-1)^{v-u} \leq (2t-1)^v.$$

Combining the above inequality with the previous upper bound,

$$|h_{x,v}v!| \leq \ell_f\left(\frac{1}{nt}\right)\frac{u_{\max}}{nt}e^{-r} \sum_{j=v+2}^{\infty} \frac{r^j}{j!}(2t-1)^v$$

$$\leq \ell_f\left(\frac{1}{nt}\right)\frac{u_{\max}}{nt}e^{-r} \sum_{j=v+2}^{\infty} \frac{((2t-1)r)^j}{j!}$$

$$\leq \ell_f\left(\frac{1}{nt}\right)\frac{u_{\max}}{nt}e^{-r} \sum_{j=0}^{\infty} \frac{((2t-1)r)^j}{j!}$$

$$= \ell_f\left(\frac{1}{nt}\right)\frac{u_{\max}}{nt}e^{2r(t-1)},$$

where the last equality follows from the Taylor expansion of $e^y$. $\qquad\square$

The above results yield the following upper bound on $\mathrm{Var}(f_S^*)$.

**Lemma 18.** *For the set of parameters specified in Section 2, if* $c_1\sqrt{c_2} \leq 1/11$ *and* $t > 2.5$, *then*

$$\mathrm{Var}(f_S^*) \leq \left(1 \wedge \frac{k}{n}\right)\frac{9s_0^2}{n^{0.22}}\ell_f^2\left(\frac{1}{nt}\right).$$

*Proof.* By Lemma 16 and Lemma 17,

$$\mathrm{Var}(f_S^*) \le (n \wedge k) \left( \ell_f \left( \frac{1}{nt} \right) \frac{u_{\max}}{nt} \right)^2 e^{4r(t-1)}.$$

Note that $t > 2.5$,

$$\frac{u_{\max}}{nt} = \frac{2s_0 t + 2s_0 - 1}{nt} \le \frac{2s_0 t + 2s_0}{nt} \le \frac{3s_0}{n},$$

and since $c_1 \sqrt{c_2} \le 0.1$,

$$4r(t-1) = 40s_0(t+1)(t-1) \le 94s_0(t-1)^2 = 94c_1^2 c_2 \log n \le 0.78 \log n.$$

Hence,

$$\left( \ell_f \left( \frac{1}{nt} \right) \frac{u_{\max}}{nt} \right)^2 e^{4r(t-1)} \le \left( \frac{3s_0}{n} \right)^2 \ell_f^2 \left( \frac{1}{nt} \right) n^{0.78} \le \frac{1}{n} \frac{9s_0^2}{n^{0.22}} \ell_f^2 \left( \frac{1}{nt} \right),$$

which implies that

$$\mathrm{Var}(f_S^*) \le \left( 1 \wedge \frac{k}{n} \right) \frac{9s_0^2}{n^{0.22}} \ell_f^2 \left( \frac{1}{nt} \right).$$

$\square$

## 8.4 Bounding the Bias of $f_S^*$

Recall that

$$\begin{aligned}
\mathrm{Bias}(f_S^*) &= \mathbb{E}[f_S^*(X^N, X^{N'}) - K_f] \\
&= \mathbb{E}\Big[\sum_{x \in \mathcal{X}} h_{x,N_x} \cdot N_x! \cdot \mathbb{1}_{N_x' \le s_0} - \sum_{x \in \mathcal{X}} h_x(\lambda_x) \mathbb{E}[\mathbb{1}_{N_x \le s_0}]\Big] \\
&= \sum_{x \in \mathcal{X}} (\hat{h}_x(\lambda_x) - h_x(\lambda_x)) \mathbb{E}[\mathbb{1}_{N_x \le s_0}],
\end{aligned}$$

which yields

$$\begin{aligned}
|\mathrm{Bias}(f_S^*)| &\le \sum_{x \in \mathcal{X}} \left| \hat{h}_x(\lambda_x) - h_x(\lambda_x) \right| \\
&= \sum_{x \in \mathcal{X}} \left| \sum_{u=1}^{u_{\max}} \frac{g_x(u)}{u!} \left( \int_r^\infty e^{-\alpha} \alpha^u f_u(\alpha \lambda_x (t-1)) d\alpha \right) \right|
\end{aligned}$$

The following lemma bounds $|f_u(y)|$ by simple functions and allows us to deal with the integral.

**Lemma 19.** *For $u \ge 1$ and $y \ge 0$,*

$$|f_u(y)| \le 1 \wedge \frac{y}{u+1}.$$

*Proof.* For $u \ge 1$ and $y \ge 0$, we have the following well-known upper bound [26] for the Bessel function of the first kind.

$$J_u(y) \le 1 \wedge \frac{(y/2)^u}{u!},$$

which implies

$$f_u(y) = J_{2u}(2\sqrt{y}) \le 1 \wedge \frac{(y)^u}{(2u)!}.$$

If $y \ge u + 1$, then

$$f_u(y) \le 1 \wedge \frac{(y)^u}{(2u)!} \le 1 \le \frac{y}{u+1}.$$

If $u + 1 > y \ge 0$, then

$$f_u(y) \le 1 \wedge \frac{(y)^u}{(2u)!} \le \frac{(y)^u}{(2u)!} \le \frac{(u+1)^u}{(2u)!} \frac{y}{u+1} \le \frac{y}{u+1} \le 1.$$

$\square$

To bound $|\text{Bias}(f_S^*)|$, it suffices to bound $|\hat{h}_x(\lambda_x) - h_x(\lambda_x)|$. The lemma below follows from the first half of Lemma 19, i.e., $|f_u(y)| \le y/(u+1)$.

**Lemma 20.** *For $t > 2.5$ and $s_0 \ge 1$,*

$$|\hat{h}_x(\lambda_x) - h_x(\lambda_x)| \le \frac{\lambda_x}{n}\ell_f\left(\frac{1}{nt}\right)e^{-2s_0 t}.$$

*Proof.* Since $|f_u(y)| \le y/(u+1)$,

$$|\hat{h}_x(\lambda_x) - h_x(\lambda_x)| \le \sum_{u=1}^{u_{\max}} \frac{|g_x(u)|}{(u+1)!}y(t-1)\int_r^\infty e^{-\alpha}\alpha^{u+1}d\alpha.$$

Note that the integral is actually the incomplete Gamma function, we can rewrite the last term as

$$\lambda_x(t-1)\sum_{u=1}^{u_{\max}} \frac{|g_x(u)|}{(u+1)!}(u+1)!e^{-r}\sum_{i=0}^{u+1}\frac{r^i}{i!} = \lambda_x(t-1)\sum_{u=1}^{u_{\max}} |g_x(u)|e^{-r}\sum_{i=0}^{u+1}\frac{r^i}{i!}.$$

Consider each term in the summation, by Lemma 2, $r = 10s_0 t + 10s_0$, and $u_{\max} = 2s_0 t + 2s_0 - 1$, for $1 \le u \le u_{\max}$,

$$|g_x(u)|e^{-r}\sum_{i=0}^{u+1}\frac{r^i}{i!} = \left(\frac{t}{t-1}\right)^u \text{Pr}(\text{Poi}(r) \le u+1)\left|f\left(\frac{u}{nt}\right)\right|$$

$$\le \left(\frac{t}{t-1}\right)^u \text{Pr}(\text{Poi}(r) \le 2s_0 t + 2s_0)\frac{3s_0}{n}\ell_f\left(\frac{1}{nt}\right)$$

$$\le \left(\frac{t}{t-1}\right)^u e^{-4.78(s_0 t + s_0)}\frac{3s_0}{n}\ell_f\left(\frac{1}{nt}\right).$$

Hence,

$$\lambda_x(t-1)\sum_{u=1}^{u_{\max}} |g_x(u)|e^{-r}\sum_{i=0}^{u+1}\frac{r^i}{i!} \le \lambda_x(t-1)e^{-4.78(s_0 t + s_0)}\frac{3s_0}{n}\ell_f\left(\frac{1}{nt}\right)\sum_{u=1}^{u_{\max}}\left(\frac{t}{t-1}\right)^u$$

$$\le \frac{\lambda_x}{n}\ell_f\left(\frac{1}{nt}\right)\left((t-1)^2 3s_0\right)e^{-4.78(s_0 t + s_0)}\left(\frac{t}{t-1}\right)^{2s_0 t + 2s_0}.$$

Note that $t > 2.5$ yields $\frac{t}{t-1} \le e^{0.64}$ and thus

$$e^{-4.78(s_0 t + s_0)}\left(\frac{t}{t-1}\right)^{2s_0 t + 2s_0} \le e^{-4.78(s_0 t + s_0)}e^{1.28(s_0 t + s_0)}$$

$$= e^{-3.5(s_0 t + s_0)}.$$

Furthermore,

$$\left((t-1)^2 3s_0\right)e^{-3.5(s_0 t + s_0)} = \left(e^{-1.5s_0 t}(t-1)^2\right)\left(e^{-3.5s_0}3s_0\right)e^{-2s_0 t}$$

$$\le e^{-2s_0 t},$$

which completes the proof. $\qquad\square$

Analogously, applying the second half of Lemma 19, i.e., $|f_u(y)| \le 1$, we get the following alternative upper bound.

**Lemma 21.** *For $t > 2.5$ and $s_0 \ge 1$,*

$$|\hat{h}_x(\lambda_x) - h_x(\lambda_x)| \le \frac{1}{n}\ell_f\left(\frac{1}{nt}\right)e^{-2s_0 t}.$$

Lemma 20 and Lemma 21 together yield the following upper bound.

**Lemma 22.** *For $t > 2.5$ and $s_0 \ge 1$,*

$$\text{Bias}(f_S^*)^2 \le \left(1 \wedge \frac{k^2}{n^2}\right)e^{-4s_0 t}\ell_f^2\left(\frac{1}{nt}\right).$$

## 8.5 Bounding $\mathbb{E}[C^2]$

Combining all the previous results, for the set of parameters specified in Section 2, if $c_1\sqrt{c_2} \leq 1/11$, $t > 2.5$, $n \geq 150$, and $1 \leq s_0 \leq \log^{0.2} n$,

$$
\begin{aligned}
\mathbb{E}[C^2] &\leq \operatorname{Var}(f_S^*) + (1 + \log n)\operatorname{Bias}(f_S^*)^2 + \left(1 + \frac{1}{\log n}\right) R_f^2 \\
&\leq \left(1 \wedge \frac{k}{n}\right) \frac{9s_0^2}{n^{0.22}} \ell_f^2\left(\frac{1}{nt}\right) + (1 + \log n)\left(1 \wedge \frac{k^2}{n^2}\right) e^{-4s_0 t} \ell_f^2\left(\frac{1}{nt}\right) \\
&\quad + \left(1 + \frac{1}{\log n}\right)\left(\left(7.1\left(1 \wedge \frac{k}{n}\right)\ell_f\left(\frac{1}{n}\right) e^{-0.3 s_0} \log n\right)^2 + \left(\frac{7.1}{n^{3.8}}\ell_f\left(\frac{1}{n}\right)\right)^2\right) \\
&\leq 8^2\left(1 \wedge \frac{k}{n}\right)\ell_f^2\left(\frac{1}{nt}\right)\log^2 n\left(\frac{1}{e^{0.6 s_0}} + \frac{1}{n^{0.22}}\right) + \left(\frac{8}{n^{3.8}}\ell_f\left(\frac{1}{n}\right)\right)^2 \\
&\leq 13^2\left(1 \wedge \frac{k}{n}\right)\ell_f^2\left(\frac{1}{nt}\right)\left(\frac{\log^2 n}{e^{0.6 s_0}}\right)
\end{aligned}
$$

## 9 Main Results

To summarize, for properly chosen parameters and sufficiently large $n$,

$$
\mathbb{E}[A^2] \leq \frac{1 + T(n)}{nt}\ell_f^2\left(\frac{1}{nt}\right) + \left(1 + \frac{1}{T(n)}\right) L_{f^E}(p, nt),
$$

$$
\mathbb{E}[B^2] \leq (8S_f)^2\left(\frac{1}{s_0} \wedge \frac{k}{n}\right) + 10\ell_f^2\left(\frac{1}{nt}\right)\frac{s_0}{n},
$$

and

$$
\mathbb{E}[C^2] \leq 13^2\left(1 \wedge \frac{k}{n}\right)\ell_f^2\left(\frac{1}{nt}\right)\left(\frac{\log^2 n}{e^{0.6 s_0}}\right),
$$

where $T$ is an arbitrary positive function over $\mathbb{N}$. Furthermore, Cauchy-Schwarz inequality implies

$$
(f^*(X^N, X^{N'}) - f(p))^2 = (A + B + C)^2 \leq (T(n)(C + B)^2 + A^2)\left(1 + \frac{1}{T(n)}\right).
$$

Choosing $T(n) = \log^\epsilon n$, the estimation loss of $f^*$ is thus bounded by

$$
\begin{aligned}
L_{f^*}(p, 2n) &= \mathbb{E}[(f^*(X^N, X^{N'}) - f(p))^2] \\
&\leq \mathbb{E}\left[(\log^\epsilon n(C + B)^2 + A^2)\left(1 + \frac{1}{\log^\epsilon n}\right)\right] \\
&\leq 2(1 + \log^\epsilon n)(\mathbb{E}[C^2] + \mathbb{E}[B^2]) + \left(1 + \frac{1}{\log^\epsilon n}\right)\mathbb{E}[A^2] \\
&\leq 2(1 + \log^\epsilon n)\left(\mathbb{E}[C^2] + \mathbb{E}[B^2] + \frac{1 + \log^\epsilon n}{2nt\log^\epsilon n}\ell_f^2\left(\frac{1}{nt}\right)\right) \\
&\quad + \left(1 + \frac{1}{\log^\epsilon n}\right) L_{f^E}(p, nt).
\end{aligned}
$$

For any property $f$ and set of parameters that satisfy the assumptions in Section 2,

$$
\mathbb{E}[C^2] + \mathbb{E}[B^2] + \frac{1 + \log^\epsilon n}{2nt\log^\epsilon n}\ell_f^2\left(\frac{1}{nt}\right) \leq C_f' \min\left\{\frac{k}{n} + \tilde{\mathcal{O}}\left(\frac{1}{n}\right), \frac{1}{\log^{2\epsilon} n}\right\},
$$

where $C_f'$ is a fixed constant that only depends on $f$.

Setting $c_1 = 1$ yields Theorem 1 with $C_f = 4C_f'$.

In Theorem 1, for fixed $n$, as $\epsilon \to 0$, the final slack term $1/\log^{\epsilon} n$ approaches a constant. For certain properties it can be improved. For normalized support size, normalized support coverage, and distance to uniformity, a more involved estimator improves this term to

$$C_{f,\gamma} \min \left\{ \frac{k}{n \log^{1-\epsilon} n} + \frac{1}{n^{1-\gamma}}, \frac{1}{\log^{1+\epsilon} n} \right\},$$

for any fixed constant $\gamma \in (0, 1/2)$.

For Shannon entropy, correcting the bias of $f_L^*$ and further dividing the probability regions, reduces the slack term even more, to

$$C_{f,\gamma} \min \left\{ \frac{k^2}{n^2 \log^{2-\epsilon} n} + \frac{1}{n^{1-\gamma}}, \frac{1}{\log^{2+2\epsilon} n} \right\}.$$

## 10   Experiments

We demonstrate the new estimator's efficacy by applying it to several properties and distributions, and comparing its performance to that of several recent estimators [13–15, 22, 27]. As outlined below, the new estimator was essentially the best in almost all experiments. It was out-performed, essentially only by PML, and only when the distribution is close to uniform.

### 10.1   Preliminaries

We tested five of the properties outlined in the introduction section: Shannon entropy, normalized support size, normalized support coverage, power sums or equivalently Rényi entropy, and distance to uniformity. For each of the five properties, we tested the estimator on the following six distributions. a distribution randomly generated from Dirichlet prior with parameter 2; uniform distribution; Binomial distribution with success probability 0.3; geometric distribution with success probability 0.99; Poisson distribution with mean 3,000; Zipf distribution with power 1.5. All distributions had support size $k = 10,000$. The Geometric, Poisson, and Zipf distributions were truncated at $k$ and re-normalized. Note that the parameters of the Geometric and Poisson distributions were chosen so that the expected value would be fairly large.

We compared the estimator's performance with $n$ samples to that of four other recent estimators as well as the empirical estimator with $n$, $n\sqrt{\log n}$, and $n \log n$ samples.

The graphs denotes NEW by $f^*$, $f^E$ with $n$ samples by Empirical, $f^E$ with $n\sqrt{\log n}$ samples by Empirical+, $f^E$ with $n \log n$ samples by Empirical++, the pattern maximum likelihood estimator in [15] by PML, the Shannon-entropy estimator in [27] by JVHW, the normalized-support-size estimator in [14] and the entropy estimator in [13] by WY, and the smoothed Good-Toulmin Estimator for normalized support coverage estimation [22], slightly modified to account for previously-observed elements that may appear in the subsequent sample, by SGT.

While the empirical estimator and the new estimator have the same form for all properties, as noted in the introduction, the recent estimators are property-specific, and each was derived for a subset of the properties. In the experiments we applied these estimators to the properties for which they were derived. Also, additional estimators [28–34] for various properties were compared in [13, 14, 22, 27] and found to perform similarly to or worse than recent estimators, hence we do not test them here.

As outlined in Section 1, the new estimator $f^*$ uses two key parameters $t$ and $s_0$ that determine and all other parameters. To avoid over-fitting, the data sets used to determine $t$ and $s_0$ was disjoint from the one used to generate the results shown.

Table 2: Values of $t$ and $s_0$ for different properties

| Property | $t$ | $s_0$ |
|---|---|---|
| Shannon Entropy | $2\log^{0.8} n + 1$ | $16\log^{0.2} n$ |
| Normalized Support Size | $\log^{0.7} n + 1$ | $16\log^{0.2} n$ |
| Normalized Support Coverage | $\log^{0.8} n + 1$ | $8\log^{0.2} n$ |
| Power Sum (0.75) | $\log^{1.0} n + 1$ | $4\log^{0.2} n$ |
| Distance to Uniformity | $\log^{0.7} n + 1$ | $4\log^{0.2} n$ |

Due to the nature of our worst-case analysis and the universality of our results over all possible distributions, we only proved that $f^*$ with $n$ samples works as well as $f^E$ with $n\sqrt{\log n}$ samples. In practice, we chose the amplification parameter $t$ as $\log^{1-\alpha} n + 1$, where $\alpha \in \{0.0, 0.1, 0.2, ..., 0.6\}$ was selected based on independent data, and similarly for $s_0$. Since $f^*$ performed even better than Theorem 1 guarantees, $\alpha$ ended up between 0 and 0.3 for all properties, indicating amplification even beyond $n\sqrt{\log n}$. Finally, to compensate the increase of $t$, in the computation of each coefficient $h_{x,v}$ we substituted $t$ by $\max\{t/1.5^{v-1}, 1.5\}$.

## 10.2 Experimental Results

With the exception of normalized support coverage, all other properties were tested on distributions of support size $k = 10,000$ and number of samples, $n$, ranging from 1,000 to 100,000. Each experiment was repeated 100 times and the reported results reflect their mean squared error (MSE). The distributions shown in the graphs below are arranged in decreasing order of uniformity. In all graphs, the vertical axis is the MSE over the 100 experiments, and the horizontal axis is $\log(n)$.

### Shannon Entropy

For the Dirichlet-drawn and uniform distributions, all recent estimators outperformed the empirical estimator, even when it was used with $n \log n$ samples. The best estimator depended on the distribution, but the new estimator $f^*$ performed best or essentially as well as the best for all six distributions.

Figure 1: Shannon Entropy

## Normalized Support Size

For the Dirichlet-drawn and uniform distributions, PML and the empirical estimators were best for small $n$, with the new estimator next. For the remaining four distributions, empirical with $n \log n$ samples was best, but among all estimators using $n$ samples and even empirical with $n\sqrt{\log n}$ samples, the new estimator was best.

Figure 2: Normalized Support Size

## Normalized Support Coverage

For this property the parameter $m$ was set to $5,000$. All the distributions have support size $k = 1,000$ and $n$, the number of samples, ranges from $1,000$ to $3,000$. The new estimator was essentially best for all distributions.

Figure 3: Normalized Support Coverage

**Power Sum (0.75), or equivalently Rényi entropy with parameter 0.75**

Again PML was best for the Dirichlet-drawn and uniform distributions, however, its performance was not as stable as $f^*$. The new estimator performed as well as $f^E$ with $n\sqrt{\log n}$ samples in all cases and matched $f^E$ with $n\log n$ samples for half of the distributions.

Figure 4: Power Sum (0.75)

## Distance to Uniformity

The new estimator performed as well as $f^E$ with $n \log n$ samples in all cases. PML was the best estimator for the Dirichlet-drawn and uniform distributions, but provided no improvement over the $n$-sample empirical estimator for half of the distributions.

Figure 5: Distance to Uniformity