[Reviews · NeurIPS 2018]

Reviewer 1



This paper considers estimation of functionals of distributions in the high-dimensional regime, and designs an estimator that “amplifies” the amount of data required to attain the same error as compared to the plug-in approach. In particular, the least squares error of functional estimation employing order n\sqrt{\log n} samples to compute the plug-in estimator can be matched by their estimator in just order n samples. The estimator uses a variant of the plug-in estimator for higher probabilities, and a polynomial smoothing technique for low probabilities, with a data-dependent threshold to decide whether a particular symbol probability is low or high. The estimator runs in inear-time in number of samples, is competitive with the plug-in estimator with a larger number of samples, simple to compute, and sees improvement in simulations. The rates improve upon the plug-in estimator in the high dimensional regime where the support size is greater than the number of samples. The paper is well-written and easy to follow, and I liked reading it. However, I did have a few concerns with some parts of it, in particular about how significant such an amplification result really is. These are listed in detail below: Concerns: 1. As the authors mention is Section 4, Theorem 1 only really provides improvement over the least squares estimate when the loss of empirical (plug-in) estimator is slowly decaying ( log^{-\eps} n). In particular, this means that even with data amplification and substituting n for n\sqrt{n}, the rate of the improved estimator can only be improved in the second (double logarithmic) term. While such slow rates are understandable in high dimensions, the afforded improvement seems miniscule here. 2. As a counter to the point above, the authors show that for some functionals, the rate of the plug-in estimate beyond which one gains improvement can actually be as low as log^{-2} n. In this case, too, it seems like the amplification afforded would only be doubly logarithmic in the rate. 3. While the two points are not major weaknesses (the simulations seem to address whether or not an improvement is actually seen in practice), the paper would benefit from a discussion of the fact that the targeted improvement is in the (relatively) small n regime. 4. The paper would benefit from a more detailed comparison with related work, in particular making a detailed comparison to the time complexity and competitiveness of prior art. Minor: 1. The proofs repeatedly refer to the Cauchy inequality, but it might be better given audience familiarity to refer to it as the Cauchy-Schwarz inequality. Post-rebuttal: I have read the authors' response and am satisfied with it. I maintain my vote for acceptance.

Reviewer 2



The authors address the problems of sample-efficient estimation of properties of discrete distributions. After identifying the shortcomings of the current state of the art (most estimators are property-specific, general ones usually have worse than linear running time, minmax estimators being too "pesimistic") they propose a new estimator that alleviates those shortcomings, applying to all Lipschitz properties in linear time. I'm not an expert in this subfield, but it looks like a solid piece of work with potentially big implications. Comments: * Are there any properties that are frequently used and do not satisfy the Lipschitz condition in Sec. 5? * I'd suggest introducing the estimator earlier on in the paper, while still deferring the derivation to a later section. At the very least, present the full form of the estimator at a prominent point, since now it's up to the reader to find the different pieces in Sec. 6 and tie them together. * Summarizing some of the key experimental results in a table would probably help the readers * I'd encourage the authors to add a brief section with conclusions at the end of the paper

Reviewer 3



This paper provides an estimator for estimating a class of functionals of some discrete probability mass functions with some favorable properties. Specifically, the proposed estimators have linear time complexity. In terms of sample complexity, they compare well with the straightforward empirical estimator computed with O(n \sqrt{log n}) samples. This paper address an interesting problem and provides an interesting solution. The novelty content paper is also high and the work is thorough. The intuition provided in section 6 is useful and the paper is overall well-written. One question I have is can lower bounds be proved for the upper bounds in theorem 1 ? If so, that would help develop more insight into this estimator. note: I have not checked the proofs line-by-line as there is not much time to review all allotted papers. But the general strategy looks ok to me.